# The Effects of Slaughter Age and Sex on Carcass Traits, Meat Quality, and Leg Bone Characteristics of Farmed Common Pheasants (*Phasianus colchicus* L.)

**DOI:** 10.3390/ani14071050

**Published:** 2024-03-29

**Authors:** Dariusz Kokoszyński, Joanna Żochowska-Kujawska, Marek Kotowicz, Hubert Piątek, Karol Włodarczyk, Henrieta Arpášová, Barbara Biesiada-Drzazga, Marcin Wegner, Mohamed Saleh, Maciej Imański

**Affiliations:** 1Department of Animal Breeding and Nutrition, Bydgoszcz University of Science and Technology, 85084 Bydgoszcz, Poland; hubertpiatek555@gmail.com; 2Department of Meat Science, West Pomeranian University of Technology, 71550 Szczecin, Poland; joanna.zochowska-kujawska@zut.edu.pl (J.Ż.-K.); marek.kotowicz@zut.edu.pl (M.K.); 3Institute of Agricultural and Food Biotechnology–State Research Institute, 02532 Warsaw, Poland; karwlo19@gmail.com; 4Institute of Animal Husbandry, Slovak University of Agriculture, 94976 Nitra, Slovakia; 5Institute of Animal Science and Fisheries, Siedlce University of Natural Science and Humanities, 08110 Siedlce, Poland; barbaradrzazga@wp.pl; 6Boehringer-Ingelheim, 00728 Warsaw, Poland; marcin.wegner@op.pl; 7Department of Poultry Science, Sohag University, Sohag 82524, Egypt; bydg2016@gmail.com; 8Animal Breeding Center, Polish Hunting Association, 88150 Rożniaty, Poland; ohz.rozniaty@pzlow.pl

**Keywords:** pheasants, sex, age of slaughter, meat quality

## Abstract

**Simple Summary:**

Pheasants are medium-sized birds from Asia and southeastern Europe that were introduced to many countries around the world. In Poland, they are mainly used as game birds. The farm rearing of pheasants for meat production is small in scale. They are also kept in parks and gardens. Due to their attractive appearance, including multi-colored plumage, especially in males, they are popular birds and are often kept by hobbyists. The aim of this study was to determine the effects of slaughter age (the duration of the rearing period) and sex on the carcass weight and composition, physicochemical properties, meat texture, and biometric traits of the leg bones (femur and tibia) of common pheasants kept on farms. The results indicate that the sex of the birds has a greater influence on these traits than the slaughter age. The numerous differences in the carcass composition, basic chemical composition, texture, and physicochemical characteristics of the meat of male and female pheasants indicate that they differ in their nutritional value and suitability for processing.

**Abstract:**

The aim of this study was to determine the effects of slaughter age and sex on the carcass characteristics, meat quality parameters, and leg bone dimensions of common pheasants. The study material consisted of 40 common pheasants, including 10 males and 10 females at 12 weeks of age and 10 males and 10 females at 15 weeks of age. The birds were kept on a farm in a semi-intensive system. The slaughter age had significant effects on the eviscerated carcass weight; the percentage of skin with subcutaneous fat; the wings percentage; electrical conductivity (EC_24_); the work required to cut the samples (cutting work); redness (a*); the intramuscular fat content in the breast meat; and the protein, intramuscular fat, water, and collagen contents in the leg meat. The sex of the pheasants had significant effects on the eviscerated carcass weight; the contents of leg muscles, skin with subcutaneous fat, and wings in the carcasses; and the electrical conductivity, thermal loss, lightness (L*), and redness (a*) of the breast muscles. It also significantly affected the protein and water contents of the pectoral and leg muscles, the intramuscular fat contents of the leg muscles, the texture traits of the pectoralis major muscle, and all femur and tibia bone dimensions. The results indicate a greater influence of sex compared to slaughter age on the pheasant traits studied. We confirmed the occurrence of a clearly marked sexual dimorphism in birds of this species. Both factors (slaughter age and gender) have significant effects on the nutritional and technological value of pheasant meat. The few studies on meat texture and the dimensions of pheasant leg bones indicate a need for continued research in this area in the future.

## 1. Introduction

The main purpose of pheasant farming in Poland is to rear large numbers of birds for the introduction of new individuals into the wild. The relatively high nutritional value of pheasant meat and its culinary suitability are not without significance. In Poland, pheasant carcasses used for meat are sourced from hunting or rearing. In both cases, the result is a high-quality product of increasing interest to consumers looking for new safe foods [1,2].

The high nutritional and dietary value of pheasant meat is indicated by its high content of complete protein and low amount of fat. Game pheasant breast meat contains 25.4% to 27.2% crude protein and 0.8% to 1.3% intramuscular fat, while the leg meat contains 22.2% to 26.3% CP and 1.8% to 4.3% fat [3,4]. According to the USDA Foods Database [5], 100 g of pheasant breast meat contains 133 kcal, while leg meat contains 134 kcal. As reported by Vitula et al. [6], the energy content of pheasant breast meat is higher than that of guinea fowl and chukar breast meat but lower than that of grey partridge and Japanese quail breast meat. The main essential amino acids in pheasant meat are lysine and leucine, while the predominant non-essential amino acids are glutamic acid and aspartic acids [4,7,8]. The content of unsaturated fatty acids in pheasant meat lipids exceeds 50% [3,4,9,10]. Pheasant meat also has low cholesterol content [2] in the breast muscles (29–48 mg/100 g of meat) and leg muscles (67–71 mg/100 g), similar to that in broiler chicken meat [11,12,13]. Pheasant meat is also a source of minerals and vitamins and has good sensory qualities, which are also of great importance to consumers of this type of meat [13,14].

Another important quality characteristic of meat is its texture. The texture of meat affects consumer acceptance, the eating experience, and satisfaction. Characteristics such as tenderness, juiciness, and chewiness have strong influences on the consumer perception of meat as a food product [15]. Meat texture depends on the type and structure of the muscle fibers, the proportion and quality of the connective tissue, the fat content, and the rate of postmortem proteolytic changes [16]. Balowski et al. [17], summarizing the results of experiments by many researchers, concluded that meat texture is influenced by direct factors, such as the species, age, and sex of the animal and the type of muscle, as well as indirect factors, such as the rearing and slaughtering conditions, cooling, aging, and the method of heat treatment.

Young pheasants reared for slaughter are raised to 10–16 weeks of age to obtain lightweight carcasses that are well suited to rapid heat treatments, such as grilling, or to 20–24 weeks of age, when the carcasses are intended for roasting [18]. According to Yamak et al. [19], pheasants reach their most suitable slaughter age for meat at 16 weeks. In contrast, Adamski and Kuźniacka [20] indicate that the rearing of young slaughter pheasants should last from 16 to 20 weeks under an intensive feeding regime.

Slaughter age and sex significantly modify carcass characteristics and meat quality parameters. Previous studies have determined the effects of the slaughter age and sex of pheasants on carcass weight and composition [20,21,22,23,24,25], the basic chemical composition of the meat [25], meat acidity and color parameters [2,12,25], the fatty acid profile, meat sensory traits [13], and meat tenderness [26].

In this study, the dimensions of the femur and tibia were also determined. For pheasants bred for introduction into the wild, anatomopathological bone characteristics, including those of leg bones, may partly determine their chances of survival. A high rate of weight gain as a result of an excessive concentration of nutrients in the diet or excessive feed intake by pheasants can lead to extreme stress on the femur and tibia, the occurrence of various diseases or pathological changes in the locomotor system, and restricted movement. In pheasants introduced into the wild, this reduces their chances of escaping predators and surviving. After the period of intensive growth in young pheasants, it is necessary to introduce a diet with a lower nutrient concentration, containing whole grains and other feedstuffs found in the wild, and to provide aviaries so that the pheasants are better prepared for life in the wild [2,13].

The authors are not aware of any studies in the available literature on the effects of pheasant age on the meat texture traits studied here (hardness, chewiness, springiness, cohesiveness, and gumminess), meat electrical conductivity, or tibial and femoral dimensions.

The aim of this study was to determine the effects of slaughter age and sex on the carcass weight and composition, the basic chemical composition and physicochemical characteristics of the breast and leg meat, the texture of the pectoralis major muscle, and the dimensions of the leg bones (femur and tibia) of common pheasants. The research hypothesis assumed the influence of slaughter age and sex on the carcass and meat quality characteristics and leg bone traits of the examined farmed common pheasants.

## 2. Materials and Methods

### 2.1. Study Material

A total of 40 common pheasants (*Phasianus colchicus* L.), including 10 males and 10 females at the age of 12 weeks and 10 males and 10 females at the age of 15 weeks, were used in this study. The study material was obtained at the Animal Breeding Centre belonging to the Management Board of the Polish Hunting Association in Rożniaty near Kruszwica (Poland). This study was performed with the permission of the Local Ethics Committee for Research on Animals in Bydgoszcz (Resolution No. 21/2014 of 26 June 2014).

### 2.2. Maintenance Conditions and Feeding of Pheasants

According to the breeder, up to and including week 3, the birds were housed in a 50 m^2^ enclosed room with regulated environmental conditions, on a concrete floor covered with sawdust. On the first day after hatching, the pheasants’ living zone was kept at 35.0 °C. On days of rearing, the temperature in the living zone of the pheasants was reduced by 1 °C. The temperature was 26–28 °C on days 8 to 14 and 21–25 °C on days 15 to 21. Infrared heaters were used as heat and light sources. The relative humidity of the air was 55–60% until day 21. From 4 to 9 weeks of age (inclusive), the pheasants were housed in a small 300 m^2^ aviary (30 m × 10 m) at a stocking rate of 3 birds/m^2^. In the last phase of rearing, from 10 to 15 weeks of age, the pheasants were kept in two large aviaries, called winter aviaries. Each winter aviary was 6400 m^2^ (80 m × 80 m). The males were kept in one aviary, and the females were kept in the other. Sex identification was based on the color of the feathers at nine weeks of age, determined during the transfer of the birds from the small aviary to the winter aviaries. Both aviaries were overgrown with spring barley and rapeseed. The substrate in both aviaries was sand. From 4 to 15 weeks of age, when the pheasants were housed in the aviaries, the environmental parameters were those of the external environment.

During the rearing period, the pheasants were fed ad libitum with complete industrial feed for slaughter pheasants. Until 3 weeks of age, the pheasants were fed Pheasant 1 ground feed. From 4 to 12 weeks of age, they received Pheasant 2 granular feed. In the final stage of rearing, from 13 to 15 weeks of age, the birds were fed Pheasant 3 granular feed and additionally received whole maize grain (60 g/bird/day). Data on the basic chemical compositions of these diets and their contents of selected amino acids and energy values are shown in Table 1. These diets contained wheat, maize, soybean meal, rapeseed meal, calcium carbonate, monocalcium phosphate, soybean oil, and sodium chloride. Furthermore, the Pheasant 1 and Pheasant 2 diets contained coccidiostats with a 5-day withdrawal period. The soybean meal was obtained from genetically modified seeds. Throughout the rearing period, the pheasants were provided unlimited access to drinking water.

### 2.3. Evaluation of Carcass Characteristics

At 12 and 15 weeks of age, 10 males and 10 females were randomly selected for slaughter, for a total of 40 pheasants from the approximately 900 birds kept in the aviary at the Pheasant Breeding Center in Rożniaty, near Kruszwica, Poland. The birds were obtained from several remote locations in each aviary (the males and females were kept in separate winter aviaries from week 9 onwards).

The pheasants were manually slaughtered by stunning with a club and were bled through ventral cuts to their neck blood vessels. This was followed by scalding, feather removal, and the evisceration of the carcasses. The eviscerated carcasses with necks were chilled in a refrigerated cabinet (18 h at 2 °C). Once the weight of each carcass was determined, they were dissected using the simplified method given by Ziołecki and Doruchowski [27]. The elements separated during the carcass dissections were weighed individually on an electronic scale; following this, the percentages of the individual carcass elements in relation to the weights of the eviscerated carcasses with necks, after chilling, were calculated.

### 2.4. Assessment of Physicochemical Characteristics

The acidity (pH_24 h_) and electrical conductivity (EC_24 h_) of the breast meat (m. pectoralis major) were determined after cooling, 24 h after slaughter. The pH of the breast meat was measured using a CX-701 Multifunction meter (Elmetron, Zabrze, Poland) equipped with a dagger electrode for determining the pH of the meat. Prior to the pH measurements, the electrode was calibrated in calibration buffers (pH 4.0 and 7.0). Using an LF-Star device from R. Matthäus, the electrical conductivity of the greater m. pectoralis major was measured by driving the electrode at an angle of 90° along the muscle fibers. The thermal loss of the breast and leg meat was determined using the method of Walczak [28]. Meat samples weighing 20 ± 1.0 g were individually wrapped in non-sterile gauze and placed in a water bath filled with water at 85 °C for 10 min. The samples were then cooled in a refrigerated cabinet for 30 min at 4 °C and weighed again. The thermal loss was calculated as the difference in weight of the meat sample before and after the heat treatment and was expressed as a percentage of the initial sample weight. The meat color parameters (CIEL*a*b*) were determined on the inner surfaces of the raw breast and leg muscles. A total of 40 breast muscle samples and 40 leg muscle samples were assessed. During the measurement, the lightness (L*), relative redness (a*) on the red–green axis, and yellowness (b*) on the yellow–blue axis were determined according to the CIE system [29]. The color scale for the L* parameter of the calibration plate was 0 for black and 100 for white. For the a* parameter, negative values indicated green and positive values indicated red, while for the b* coordinates, negative values indicated blue and positive values indicated yellow. Measurements were taken using a ChromaMeter colorimeter (Konica Minolta, Osaka, Japan) with a CR400 head at a setting for illumination compatible with a D_65_ illuminator, a 10 standard observer angle, a light projection tube with a plate with a diameter of 22 mm, and a measurement aperture with a diameter of 8 mm (and an area of 0.5 cm^2^). Measurements were taken after calibration of the instrument with a white reference tile, where Y = 86.10, x = 0.3188, and y = 0.3362. The pH_24_ measurements were repeated three times for each sample.

### 2.5. Determination of Basic Chemical Composition

The basic chemical compositions (protein, intramuscular fat, water, and collagen contents) of the breast and leg meat of the pheasants from the two slaughter age groups were determined separately using a FoodScan near-infrared spectrophotometer (FoodScan, Hillerød, Denmark) that was calibrated using artificial neural networks (ANNs) in accordance with PN-A-82109 [30]. The spectral range of reflectance was 850 to 1050 nm.

### 2.6. Determination of Texture Characteristics of the Pectoralis Major Muscle

Texture was tested on a Stable Micro Systems TA.XT plus apparatus (Stable Micro Systems, Godalming, UK) using the Texture Profile Analysis (TPA) test and the Warner–Bratzler (WB) test. Determinations were carried out in 5–6 replicates on meat samples that were heat-treated (68.8 °C at the geometric center) and then cooled to approximately 12 °C. A stylus with a diameter of 0.62 cm was used in the TPA test. Each time, a meat sample was placed under the probe and moved downwards at a constant speed of 50 mm/min, parallel to the muscle fibers, with a deformation limit of 80% of the original height (16 mm). The hardness, cohesiveness, springiness, chewiness, and gumminess were determined [31]. In the Warner–Bratzler (WB) test, meat samples with heights of 20 mm were cut parallel to the muscle fibers using a ‘V’ slot blade (HDP/WBV); the working speed of the crosshead was 50 mm/min. The WB shear force and the work required to cut the sample were determined [31].

### 2.7. Measurements of Leg Bone Dimensions

The dimensions of the femur and tibia were taken using electronic calipers according to the method given by den Driesch [32]. The following dimensions of the femur were measured: the length (greatest (GL) and medial (ML)), the smallest breadth of the corpus (SB, and the greatest breadth and greatest depth at the proximal end (GB and GD) and distal end (GC and GE, respectively). The following measurements were taken of the tibia: the length (greatest (GL) and axial (AL)), the smallest breadth of the corpus (SB), the greatest diagonal of the proximal end (GB), and the greatest breadth and depth of the distal end (SD and DD).

### 2.8. Statistical Characteristics

The data collected during this study on carcass weight and composition, basic chemical composition, physicochemical and textural traits, and femur and tibia dimensions were subjected to statistical analysis. For each trait, the arithmetic mean was calculated for both study factors (slaughter age and sex) for both groups combined. Using the Shapiro–Wilk test, an assessment was made of the correspondence of the empirical distributions of the characteristics to a normal distribution. Therefore, a parametric test in the form of a two-factor analysis of variance was used to determine the effects of slaughter age and sex on the traits. The following linear model was used: Y_ijk_ = µ + a_i_ + b_j_ + (a × b)_ij_ + e_ijk_(1)
where Y_ijk_ is the value of the trait, µ is the overall mean of the trait, a_i_ is the effect of the i_th_ slaughter age, b_j_ is the effect of the j_th_ sex, (a × b)_ij_ is the slaughter age by sex interaction, e_ijk_ is the error, and k is the k_th_ observation for the target trait in the group. 

The statistical measures of the traits were calculated using SAS software (SAS Institute, Cary, NC, USA) version 9.4. [33]. The significances of differences at *p* < 0.05 between the slaughter age groups and between males and females were verified using the Tukey post hoc test. For all traits studied in this experiment, i.e., the carcass, meat quality, and femur and tibia bone dimensions, the individual bird was the experimental unit.

## 3. Results

The average weight of the eviscerated carcasses with necks of male and female 15-week-old common pheasants was higher significantly than that of the carcasses obtained from birds at 12 weeks of age. The weights of the eviscerated carcasses with necks were significantly higher for males than for females. The older males and females had a significantly higher percentage of skin with subcutaneous fat and males a lower percentage of wings with skin. The slaughter age had no significant effect on the carcass content, i.e., breast muscles, leg muscles, neck without skin, and carcass remains. Males had significantly higher contents of leg muscles (at 12 and 15 weeks) and skin with subcutaneous fat (at 12 and 15 weeks) and a lower content of wings with skin (at 15 weeks) compared to females. For the carcass traits, there were no statistically significant interactions between the slaughter age and sex (Table 2).

Slaughter age significantly affected the electrical conductivity and redness of breast meat. Significantly higher values of EC_24_ and a* were found in older males. Males at 12 weeks of age had significantly higher thermal loss and lightness and lower electrical conductivity than females. The pectoral muscles of 15-day-old males had significantly higher EC_24_ and a* values compared to females of the same age. There were no significant interactions between slaughter age and sex for the pH_24_ and b* values of the breast meat or the L*, a*, and b* color attributes of the leg muscles (Table 3).

Pheasant age significantly affected the content of intramuscular fat in the breast meat and the contents of protein, intramuscular fat, water, and collagen in the leg meat. With age, there were significant increases in the intramuscular fat contents in the pectoral and leg muscles of the male and female pheasants. There were also significant increases in the collagen contents and reductions in the water contents in the leg muscles of males and females, as well as a reduction in the protein content in the leg muscles of males. The pectoral muscles of 12-week-old males contained significantly more protein and intramuscular fat than those of females of the same age. In contrast, there was significantly less water and intramuscular fat in the pectoral muscles of 15-week-old males than in those of females. There was more intramuscular fat and less protein and water in the leg muscles of 15-week-old males than in those of females. In addition, significant interactions between the slaughter age and sex were found for the protein, intramuscular fat, and water contents in the breast and leg muscles (Table 4).

The texture characteristics of the pectoralis major muscles of the farmed common pheasants revealed higher values for the WB shear force, the work required to cut the samples, hardness, gumminess, and chewiness in the 15-week-old pheasants than in the 12-week-old birds. However, significant differences were only found for the work required to cut the meat samples in males. Regardless of their slaughter ages, the males had significantly higher texture parameters than the females, except for the springiness of the pectoralis major muscle at the age of 12 weeks. There were no significant interactions between the slaughter age and sex for the texture traits tested (Table 5).

Increased dimensions of the femur and tibia were noted with age. However, significant differences were only found for the smallest breadth of the corpus and the greatest diagonal of the proximal end of the tibia. Males had significantly larger dimensions of the femur and tibia. There were no significant interactions between the slaughter age and sex for the femur and tibia dimensions tested (Table 6 and Table 7).

## 4. Discussion

The length of rearing of the common pheasants evaluated in this experiment significantly affected the weights of the eviscerated carcasses with necks. The carcass weight of the 15-week-old pheasants was 125.2 g higher, i.e., 20.6% higher, compared to the carcasses of pheasants aged 12 weeks. In a study by Adamski and Kuźniacka [20], the weight of eviscerated carcasses of 16-week-old pheasants was 757 g, which was 162 g (27.2%) higher than that of carcasses of 12-week-old birds. Another study [22] showed no significant differences in the weights of carcasses obtained from 13-, 14-, and 15-week-old pheasants. As in this experiment, Sarica et al. [22] and Kokoszyński et al. [23] found an effect of sex on the weights of eviscerated carcasses. Kuźniacka [3] reported that from 3 weeks of age, males have a higher body weight than females. At 16 weeks of age, the body weight of males was about 35% higher than that of females. Young common pheasants reach the size of adult birds at around 16 weeks of age, while they reach the weight of an adult bird and full plumage at 24 weeks of age. In another experiment, males were 33.1% heavier than females. The difference in body weight between males and females at 15 weeks of age was determined by Sarica et al. [22]. The results of [2] indicate that body weight is highly correlated with carcass weight.

The present evaluation showed no significant changes in the breast meat or leg meat contents of the carcasses between 12 and 15 weeks of age in the common pheasants. Adamski and Kuźniacka [20] and Yamak et al. [19] found no significant changes in the breast contents of the carcasses of pheasants between 12 and 16 weeks of age. Sarica et al. [22] also found no significant effect of slaughter age (13–15 wks) on the breast contents of pheasant carcasses. In the present study, we found non-significant increases in the percentages of breast muscles in the male and female carcasses and a significantly higher percentage of leg muscles in the male carcasses compared to the female carcasses, confirming the results of Sarica et al. [22] and Kokoszyński et al. [23]. In contrast, Adamski and Kuźniacka [20] showed no significant effect of sex on the percentages of breast and leg muscles in pheasant carcasses at 12, 16, and 20 weeks of age. Biesiada-Drzazga et al. [1] reported a non-significant effect of sex on the breast and leg muscle contents in carcasses of 20-week-old game pheasants, and Mieczkowska et al. [10] observed a similar result in the carcasses of 16-week-old birds. In our study, the length of the rearing period and sex significantly interacted, affecting the contents of skin with subcutaneous fat and wings with skin in the carcasses. Yamak et al. [19] reported a significant increase in the percentage of neck in carcasses (from 5.84% to 6.06%) between 14 and 18 weeks of age. During the same period, the wing content was reduced from 12.62% to 11.43% [20], which was statistically insignificant and contrary to the results of this experiment. Kokoszyński et al. [23] analyzed the growth of young game pheasants and found a significant effect of sex on the percentage of neck in carcasses (males, 4.5%; females, 3.9%) and a non-significant effect of age (18 weeks, 4.3%; 20 weeks, 4.2%). Regarding the percentage of neck in pheasant carcasses, no significant effect of slaughter age or sex was found [23]. In conclusion, as in previous studies, we found a relatively low content of skin with subcutaneous fat and relatively high percentages of pectoral and leg muscles in the carcasses of the pheasants, which correspond well with the requirements of the modern consumer. In addition, this indicates the relatively low energy value of pheasant carcasses. The excessive energy supply in the human diet, especially from saturated fatty acids, is considered to be one of the causes of the dramatic increase in the number of obese people, who are at increased risk of diabetes, hypertension, and heart disease [34].

One of the most important indicators of meat quality is pH. pH values affect the color, thermal loss, palatability, and juiciness of meat, as well as the stability of meat microflora [35]. In the present study, the slaughter age and sex had no significant effect on the pH_24_ of the breast muscles, which was consistent with the findings of Sarica et al. [22]. In this experiment, the pH_24_ of the breast muscles of the males (pH_24_ = 5.83) was found to be significantly lower (*p* < 0.05) than that of the females (pH_24_ = 7.52), which was probably due to the males’ greater response to pre-slaughter stress or the lower glycogen levels in the lighter females, which were characterized by greater pectoral muscle activity before slaughter. The pH_24_ values of the breast meat of the 12-week-old pheasants were higher than those of the 14-week-old pheasants (pH_24_ = 6.77) in the experiment by Sarica et al. [22]. In a study by Kuźniacka et al. [25], the pH_24_ values of the breast muscles ranged from 4.13 to 4.51, which was much lower than the values in this experiment. According to the authors of [25], the low pH_24_ of the breast muscles was related to the high stress susceptibility of pheasants and the relatively low degree of domestication of this species of commercial birds. Meat pH values can also be affected by the species of bird or the rearing conditions. Yamak et al. [36] reported significantly lower pH values in the breast and thigh muscles of partridges (*Alector chucar*) kept in a free-range (with outdoor access) system compared to birds kept in a barn (indoor system). Balowski et al. [17], on the other hand, reported a significant effect of bird species on pectoral muscle pH values. Farm-raised guinea fowl had a lower pectoral muscle pH (pH = 5.56) than farm quails (pH = 5.86) and farm partridges (pH = 5.85).

Another little-understood parameter of poultry meat quality is electrical conductivity. Regular poultry meat has low electrical conductivity immediately after the birds are slaughtered. With age, the game pheasants showed increases in their EC_24_ values, combined with an increase in meat acidity (lower pH_24_) and higher thermal loss, which should be considered undesirable traits for meat consumers [37,38]. However, to date, no electrical conductivity limits have been set to distinguish normal meat from meat with PSE (pale, soft, and exudative) or DFD (dark, firm, and dry) defects. Data on the EC_24_ breast meat values of normal pheasants are presented for the first time in this article.

Meat color is another important characteristic for the consumer and the processing industry. In this study, the older pheasants at 15 weeks of age were found to have a lighter breast and leg meat color, which was probably related to the increases in the intramuscular fat contents in both types of meat as the birds aged. As in Kuźniacka’s experiment [2], higher L* (lightness) and lower a* (redness) values were found for the breast meat compared to the leg meat. The darker color of the leg meat compared to the breast meat was probably related to a higher proportion of red fibers (with a higher hem pigment content), which are better adapted to the prolonged exertion of walking in pheasants. According to Kiessling [39], in the pheasant’s m. pectoralis major, red fibers account for 32% of the total fibers, while in the leg muscles, they account for 57%. 

The results of the basic chemical composition determinations confirmed that pheasant meat, like that of game bird species such as wild mallard duck [40], graylag goose [41], and grey partridge [42], has a high protein content and a low fat content. In this study, the slaughter age was found to have a significant effect on the intramuscular fat content of the breast meat and the water, protein, intramuscular fat, and collagen contents of the leg meat, which may indicate changes in the nutritional value of pheasant meat at successive evaluation dates. With age, there was an increase in intramuscular fat and a decrease in the water content, confirming the results of previous studies. As in other studies [4,37,38], the leg meat had higher water and fat contents than the breast meat.

The protein content of the breast meat of the common pheasants was higher than in the experiments by Hofbauer et al. [43] (25.03%) and Kuźniacka et al. [25] (24.53%), while the protein content of the leg meat was higher than in the study by Biesiada-Drzazga et al. [1] (23.3%). According to Zotte et al. [44], the lower intramuscular fat content in breast meat compared to leg meat is related to the higher percentage of white fibers in breast muscles, which have less need for energy storage. Another experiment [45] found a higher collagen content in breast meat (1.7–1.8%) compared to the results of this experiment, where the collagen content in the breast meat ranged from 1.4% to 1.5%. Daszkiewicz and Janiszewski [46] reported a much lower collagen content in pheasant breast meat at 25 weeks of age. The authors [46] further demonstrated a significantly higher collagen content in the breast meat of males (0.20%) compared to females (0.14%), which was also confirmed in this experiment. In this study, moreover, there was a significant effect of gender on the water and protein contents of the pectoral and leg muscles, as well as the intramuscular fat content of the leg muscles, which was not confirmed in the experiments by the cited authors [46]. On the other hand, Kuźniacka et al. [25] and Kokoszyński et al. [45] reported on the significant effect of sex on the protein contents of the pectoral and leg muscles of 16-week-old pheasants.

In this study, the effect of the slaughter age on texture traits (hardness, gumminess, chewiness, springiness, and cohesiveness), with the exception of meat tenderness, was determined for the first time. According to Gornowicz et al. [15], meat tenderness, expressed as the maximum shear force, and meat firmness are considered by consumers to be the most important texture characteristics of meat. The WB shear force values obtained in this study were lower (the meat was more tender) than the values obtained for the breast meat texture of 16-week-old common pheasants (males, 93.8–116.4 N; females, 80.3–92.2 N) but higher than the values reported by Augustyńska-Prejsnar et al. [47] (36.14–38.40 N). The breast meat of the females was characterized by significantly better tenderness (lower WB shear force) compared to the males, confirming the results of previous studies [9]. The inferior tenderness of the pectoralis major muscles of the male pheasants compared to the females was probably mainly related to the larger diameter of the pectoralis major muscle fibers of the males compared to the females. Balowski et al. [17], in a study of male game pheasants aged 5 months, found higher values for hardness (29.73 N), chewiness (12.65 N × cm), gumminess (11.82 N), and cohesiveness (1.11 cm) compared to the males aged 12 and 15 weeks in this experiment. Regarding the texture traits discussed above, there was a significant effect of sex in this experiment (except for springiness), which was not confirmed in earlier studies [45].

Information was only found on the lengths of the tibias of pheasants in the available literature. In an experiment by Flis et al. [48], the lengths of the tibias of 16-week-old male game pheasants ranged from 108.4 to 114.4 mm, i.e., close to the greatest length of the tibia bone of the pheasants in this study.

## 5. Conclusions

An analysis of the results obtained in this study indicates that due to the more favorable carcass composition and nutritional value of the meat, slaughtering pheasants at 12 weeks of age is more favorable for pheasant meat consumers than slaughter at 15 weeks. Pheasant rearing until 12 weeks of age also improves the economic efficiency in the production of this poultry species compared to rearing to 15 weeks of age, which is beneficial to producers of young slaughter pheasants. Due to the size and composition of their carcasses and the texture characteristics of their meat, female pheasants are better able to satisfy consumer requirements than the carcasses and meat from males. The results indicate that sex has a greater influence than the slaughter age on the pheasant traits studied. We confirmed clear sexual dimorphism in birds of this species. Until this experiment, there were no studies on the effects of the age and sex of pheasants on meat texture traits (except tenderness) or femur and tibia dimensions, which indicates the need for continued research in this area in the future. Studies on leg bones should provide information on weight and geometric characteristics, macro- and micronutrient contents, and mechanical and densitometric traits.

## Figures and Tables

**Table 1 animals-14-01050-t001:** Chemical compositions of diets for pheasants.

Ingredients	Pheasant 1	Pheasant 2	Pheasant 3
	1 to 3 Wks	4 to 12 Wks	13 to 15 Wks
Metabolizable energy (kcal/kg)	2800	2750	2750
Crude protein (%)	25.00	23.00	19.00
Crude fat (%)	3.00	3.50	3.50
Crude fiber (%)	3.50	3.50	3.50
Crude ash (%)	6.50	6.00	5.00
Calcium (%)	1.00	0.90	0.62
Phosphorus (%)	0.70	0.60	0.46
Sodium (%)	0.20	0.20	0.20
Lysine (%)	1.60	1.40	1.00
Methionine (%)	0.60	0.50	0.40
Methionine + Cystine (%)	1.00	0.95	0.75
Threonine (%)	1.00	0.95	0.75
Tyrosine (%)	0.30	0.30	0.20

**Table 2 animals-14-01050-t002:** Weights of eviscerated carcasses with necks and percentage shares of carcass elements in common pheasants.

Trait	12 Weeks		15 Weeks		SEM	*p* Values		
	Male	Female	Male	Female		A	S	A × S
Carcass weight (g)	720.44 ^b^	492.98 ^b,^*	896.89 ^a^	583.47 ^a,^*	26.740	<0.001	<0.001	0.097
Breast muscles (%)	26.13	25.08	26.47	27.79	0.343	0.074	0.519	0.196
Leg muscles (%)	21.47	19.16 *	21.34	18.54 *	0.481	0.688	0.009	0.788
Skin with fat (%)	5.64 ^b^	4.94 ^b,^*	7.08 ^a^	5.95 ^a,^*	0.178	<0.001	0.001	0.639
Wings with skin (%)	11.65 ^a^	11.56	10.42 ^b^	11.42 *	0.144	0.003	0.030	0.080
Neck without skin (%)	3.71	4.28	3.97	4.01	0.093	0.978	*0.102*	0.152
Carcass remains (%)	31.40	34.98	30.72	32.29	0.716	0.452	0.102	0.756

^a,b^ Means with different superscripts are statistically different between birds of the same sex of different ages (*p* < 0.05). * Indicates statistical differences between males and females of the same age (*p* < 0.05). n = 20/slaughter age or sex. A—slaughter age; S—sex; A × S—interaction between slaughter age and sex.

**Table 3 animals-14-01050-t003:** Physicochemical characteristics of meat of common pheasants.

Trait	12 Weeks		15 Weeks		SEM	*p* Values		
	Male	Female	Male	Female		A	S	A × S
Breast muscle								
pH_24h_	5.78	5.97	5.87	5.73	0.053	0.316	0.331	0.290
EC_24h_ (mS/cm)	10.24 ^b^	12.10 ^a,^*	12.89 ^a^	11.31 ^b,^*	0.399	<0.001	<0.001	<0.001
Thermal loss (%)	20.01	14.51 *	19.03	18.57	0.583	0.125	0.005	0.015
L*—lightness	49.63	42.45^*^	48.01	46.07	0.987	0.154	0.001	0.019
a*—redness	7.61 ^b^	7.44	12.90 ^a^	7.92 *	0.478	0.001	*0.003*	0.020
b*—yellowness	3.00	2.18	3.19	3.68	0.364	0.155	0.619	0.245
Leg muscle								
L*—lightness	40.78	37.39	41.79	41.52	0.724	0.073	0.195	0.269
a*—redness	11.15	11.79	12.10	12.11	0.518	0.755	0.561	0.571
b*—yellowness	2.41	2.94	2.31	2.90	0.319	0.739	0.286	0.850

^a,b^ Means with different superscripts are statistically different between birds of the same sex of different ages (*p* < 0.05). * Indicates statistical differences between males and females of the same age (*p* < 0.05). n = 20/slaughter age or sex. A—slaughter age; S—sex; A × S—interaction between slaughter age and sex.

**Table 4 animals-14-01050-t004:** Basic chemical compositions of meat of common pheasants.

Trait		12 Weeks		15 Weeks		SEM	*p* Values		
		Male	Female	Male	Female		A	S	A × S
Protein (%)	BM	26.98	25.98 ^b,^*	26.61	26.43 ^a^	0.079	0.745	<0.001	0.002
	LM	23.04 ^a^	22.95	22.00 ^b^	23.22 *	0.096	0.001	<0.001	<0.001
Intramuscular fat (%)	BM	0.49 ^b^	0.38 ^b,^*	1.22 ^a^	1.54 ^a,^*	0.089	<0.001	0.464	0.001
	LM	1.86 ^b^	1.52 ^b,^*	4.02 ^a^	2.11 ^a,^*	0.168	<0.001	<0.001	<0.001
Water (%)	BM	71.46	71.48	70.78	71.74 *	0.070	0.073	<0.001	<0.001
	LM	73.02 ^a^	73.17 ^a^	71.21 ^b^	72.46 ^b,^*	0.151	<0.001	0.001	0.005
Collagen (%)	BM	1.42	1.51	1.51	1.54	0.021	0.174	0.191	0.484
	LM	1.60 ^b^	1.56 ^b^	1.72 ^a^	1.80 ^a^	0.028	0.002	0.732	0.289

^a,b^ Means with different superscripts are statistically different between birds of the same sex of different ages (*p* < 0.05). * Indicates statistical differences between males and females of the same age (*p* < 0.05). n = 20/slaughter age or sex. A—slaughter age; S—sex; A × S—interaction between slaughter age and sex.

**Table 5 animals-14-01050-t005:** Texture characteristics of the pectoralis major muscles of common pheasants.

Trait	12 Weeks		15 Weeks		SEM	*p* Values		
	Male	Female	Male	Female		A	S	A × S
WB shear force (N)	72.73	47.00 *	85.97	54.45 *	3.731	0.093	<0.001	0.632
Cutting work (J)	0.57 ^b^	0.37 *	0.73 ^a^	0.45 *	0.030	0.012	<0.001	0.398
Hardness (N)	24.41	17.08 *	24.52	19.30 *	0.696	0.243	<0.001	0.292
Gumminess (N)	8.23	4.89 *	8.30	6.04 *	0.337	0.246	<0.001	0.307
Chewiness (N × cm)	11.94	7.51 *	12.14	8.77 *	0.491	0.355	<0.001	0.505
Springiness (cm)	1.45	1.51	1.47	1.44	0.017	0.482	0.655	0.228
Cohesiveness	0.33	0.28 *	0.33	0.29 *	0.006	0.246	0.004	0.275

^a,b^ Means with different superscripts are statistically different between birds of the same sex of different ages (*p* < 0.05). * Indicates statistical differences between males and females of the same age (*p* < 0.05). n = 20/slaughter age or sex. A—slaughter age; S—sex; A × S—interaction between slaughter age and sex.

**Table 6 animals-14-01050-t006:** Dimensions of the femur bones of common pheasants.

Trait	12 Weeks		15 Weeks		SEM	*p* Values		
	Male	Female	Male	Female		A	S	A × S
GL	85.36	77.65 *	86.76	77.86 *	0.856	0.234	<0.001	0.741
ML	80.14	72.85 *	80.82	72.99 *	0.770	0.242	<0.001	0.700
SB	7.30	5.99 *	7.41	6.49 *	0.174	0.409	0.001	0.277
GB	16.29	13.48 *	16.38	14.01 *	0.247	0.236	<0.001	0.531
GD	9.70	8.69 *	9.91	8.89 *	0.164	0.981	0.021	0.391
GC	14.55	12.49 *	14.81	12.26 *	0.227	0.654	<0.001	0.574
GE	10.41	8.97 *	10.42	9.37 *	0.159	0.436	<0.001	0.460

GL, greatest length; ML, medial length; SB, smallest breadth of the corpus; GB, greatest breadth of proximal end; GD, greatest depth of proximal end; GC, greatest breadth of distal end; GE, greatest depth of distal end. * Indicates statistical differences between males and females of the same age (*p* < 0.05). n = 20/slaughter age or sex. A—slaughter age; S—sex; A × S—interaction between slaughter age and sex.

**Table 7 animals-14-01050-t007:** Dimensions of the tibia bones of common pheasants.

Trait	12 Weeks		15 Weeks		SEM	*p* Values		
	Male	Female	Male	Female		A	S	A × S
GL	113.74	105.07 *	115.75	106.03 *	1.105	0.700	<0.001	0.069
AL	109.83	101.79 *	111.94	102.94 *	1.078	0.855	<0.001	0.121
SB	5.61 ^b^	5.04 *	5.96 ^a^	5.3 *	0.076	0.028	0.001	0.978
GD	13.48 ^b^	13.31 ^b^	19.46 ^a^	16.98 ^a,^*	0.478	<0.001	0.045	0.079
SD	10.72	9.44 *	10.93	9.59 *	0.184	0.371	<0.001	0.117
DD	9.38	8.05 *	9.70	8.40 *	0.283	0.482	0.015	0.780

GL, greatest length; AL, axial length; SB, smallest breadth of the corpus; GD, greatest diagonal of the proximal end; SD, smallest breadth of the distal end; DD, greatest depth of the distal end. ^a,b^ Means with different superscripts are statistically different between birds of the same sex of different ages (*p* < 0.05). * Indicates statistical differences between males and females of the same age (*p* < 0.05). n = 20/slaughter age or sex. A—slaughter age; S—sex; A × S—interaction between slaughter age and sex.

## Data Availability

Data are contained within the article.

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
