# Peer review of "The Effects of Slaughter Age and Sex on Carcass Traits, Meat Quality, and Leg Bone Characteristics of Farmed Common Pheasants (Phasianus colchicus L.)"

_animals, 2024, doi:10.3390/ani14071050_

Round 1

Reviewer 1 Report (Previous Reviewer 1)

Comments and Suggestions for Authors

Thank you for the provided response. I understand the attempt to clarify the issue, but the answers do not fully address the actual situation. The samples still had to be cut to a height of 20 mm, and this in a direction to enable the TPA test to be performed parallel to the muscle fibers. Such cutting is the most destructive to the meat structure, causing its delamination and deformation. A simple solution would be to use a probe with a larger diameter. I accept the proposed change; please consider replacing "muscle samples" with "meat samples," as the breast muscles were subjected to thermal processing before performing the TPA.

Author Response

Comments made by Reviewer 1 and response to the comments.

Quality of English Language

( ) I am not qualified to assess the quality of English in this paper
( ) English very difficult to understand/incomprehensible
( ) Extensive editing of English language required
( ) Moderate editing of English language required
( ) Minor editing of English language required
(x) English language fine. No issues detected

Comments and Suggestions for Authors

Thank you for the provided response. I understand the attempt to clarify the issue, but the answers do not fully address the actual situation. The samples still had to be cut to a height of 20 mm, and this in a direction to enable the TPA test to be performed parallel to the muscle fibers. Such cutting is the most destructive to the meat structure, causing its delamination and deformation. A simple solution would be to use a probe with a larger diameter. I accept the proposed change; please consider replacing "muscle samples" with "meat samples," as the breast muscles were subjected to thermal processing before performing the TPA.

Submission Date

22 February 2024

Date of this review

03 Mar 2024 14:48:28

Answer:

The reviewer accepted the changes made in the text regarding the method of marking texture features.

Following reviewer's comment, "muscle samples" changed to "meat samples" (L235)

Reviewer 2 Report (New Reviewer)

Comments and Suggestions for Authors

Regarding the manuscript entitled The effect of slaughter age and gender on carcass traits, meat quality and leg bone characteristics of farmed common pheasants (Phasianus colchicus L.). The manuscript needs English revision. There are many grammatical errors.

L26. Revise spelling errors

In the abstract don’t use the word ‘significant’ with the p value. Please revise also the Results section

L47. Strong conclusion should be added.

L70. Revise grammatical error.

The introduction section should focus on the main problem and clear objectives and hypothesis. Poor introduction section and it needs revisions.

Table 1. phosphorus, correct. What are the feed ingredients?

L221. Slaughter age and sex.

Conclusion section should focus on the main findings and recommendations instead of repeating the results.

Comments on the Quality of English Language

extensive

Author Response

Comments made by Reviewer 2 and response to the comments.

Comments and Suggestions for Authors

Comment 1

Regarding the manuscript entitled The effect of slaughter age and gender on carcass traits, meat quality and leg bone characteristics of farmed common pheasants (Phasianus colchicus L.). The manuscript needs English revision. There are many grammatical errors.

Answer:

After making responses to the reviewers' comments, the article was sent for revision of grammatical errors by a native speaker of MDPI English Language Editing Services recommended by the editor of Animals journal.

Comment 2

L26. Revise spelling errors

Answer

Apperancincincluding changed to „apperance including”

Comment 3

In the abstract don’t use the word ‘significant’ with the p value. Please revise also the Results section

Answer:

L40,45, 273, 300, 318, 320, 323, 343, 362, 364 deleted „(p < 0.05)”

L278,282, 365 deleted „(p > 0.05)”

L363 deleted (p = 0.028) and (p < 0.001)

Comment 4

Added

L51-54

We confirmed the occurrence of a clearly marked sexual dimorphism in birds of this species. Both factors (slaughter age and gender) have significant effects on the nutritional and technological value of pheasant meat.

Comment 5

L47. Strong conclusion should be added.

L52-53

The few studies on meat texture and the dimensions of pheasant leg bones indicate a need for continued research in this area in the future.

Comment 6

L70. Revise grammatical error.

L70

Sentence removed at the request of another Reviewer

Comment 7

The introduction section should focus on the main problem and clear objectives and hypothesis. Poor introduction section and it needs revisions.

L96-100

A description of the findings from References regarding the effect of sex and slaughter age on carcsss characteristics and meat quality is provided in the "Discussion" section.

L 110-120

Added:

In this study, the dimensions of the femur and tibia were also determined. For pheasants bred for introduction into the wild, anatomopathological bone characteristics, including those of leg bones, may partly determine their chances of survival. An excessive rate of weight gain as a result of an excessive concentration of nutrients in the diet or excessive feed intake by pheasants can lead to excessive stress on the femur and tibia, the occurrence of various diseases or pathological changes in the locomotor system, and restricted movement. In pheasants introduced into the wild, this reduces their chances of escaping predators and surviving. After the period of intensive growth in young pheasants, it is necessary to introduce a diet with a lower nutrient concentration, containing whole grains and other feedstuffs found in the wild, and to provide aviaries so that the pheasants are better prepared for life in the wild [2,13].

In a previous version of the paper, this text was placed in the "discussion" section due to a comment from one of the reviewers about the need to shorten the "Introduction" chapter

L128-130

Added research hypothesis: the following text

The research hypothesis assumed the existence of an effect of slaughter age and sex on the carcass and meat quality traits and pheasant leg bone traits studied famed common.

Comment 8

Table 1. phosphorus, correct. What are the feed ingredients?

Answer:

In table 1 „hosphorus” changed to „Phosphorus”

The component composition of the feed mixtures used in the diets of the tested pheasants is covered by trade secrets. The manufacturer of Pheasant 1, Pheasant 2 and Pheasant 3 feed mixtures - LIRA Feed Factory of Krzywiń, Poland did not agree to disclose and print the recipes of the feed mixtures.

All the information available to the authors of this article about the feeds used is included in the article.

Comment 9

L221. Slaughter age and sex.

Answer

L253

Slaughter age sex

Changed to

„slaughter age and sex”.

Comment 10

Conclusion section should focus on the main findings and recommendations instead of repeating the results.

Answer:

Deleted

L476-482 (previous version)

Older pheasants, at 15 weeks of age, had significantly higher eviscerated carcass weight and content of skin with subcutaneous fat content, with a lower percentage of wings with skin than in 12-week-old birds. Breast meat of 15-week-old pheasants had significantly higher electrical conductivity and redness than 12-week-old birds. With age, there was a significant increase in intramuscular fat content in the breast meat and leg meat and in collagen content in the leg meat, and a significant reduction in water and protein content in the leg meat.

L486-493 (previous version)

Male pheasants, compared to females,  significantly higher eviscerated carcass weight, leg meat content, and lightness (L*) and redness (a*) of breast meat, while the content of skin content with subcutaneous fat and wings and the electrical conductivity of the breast meat . The sex of the pheasants had a significant effect on the basic chemical composition, except for collagen content in the breast and leg meat and intramuscular fat content in the breast meat. In addition,  significantly higher values of texture traits of the pectoralis major muscle, except for springiness, and higher values of tibia and femur dimensions compared to females.

Reviewer 3 Report (New Reviewer)

Comments and Suggestions for Authors

The present manuscript Kokoszyński et al evaluated the effect of slaughter age and sex  on the carcass and meat quality of pheasants. Authors have conducted a good and exhaustive study but still there is a lot of scope for improvement in each section of manuscript. Thus I recommend to work more on this manuscript so to provide a better impact of the study and wide readership. The manuscript also need to be checked for grammatical errors and sentencing.

I have my observations as follows-

       i.          Simple summary: seems too much general; especially 23-26 lines

     ii.          Abstract: please be more specific for traits as some traits may be confusing at L44

   iii.          L 37: Please add further details on methodology such as 40 pheasants- plz mention the groups wise such as 20 male 20 female, age

   iv.          L40: cutting work means plz?

     v.          Keywords: Authors may follow like this

Pheasants, sex, age of slaughter, meat quality

   vi.          L56: authors may use main 1or 2 references only

  vii.          In introduction: Need to be more focussed on the objectives and hypothesis, further authors have mentioned the leg bone dimensions, but in introduction, this section is not covered.

viii.          First two paragraphs may be merged and concised as the objective of the study is to see the effect of age and sex; so extensively describing the nutritive value seems not have much relevance here.

   ix.          L79-88: sorry for my observation, but it seems for me irrelevant and may be deleted

     x.          L100-101: please correct the sentence also may add lack of studies or scarcity of published studies etc

   xi.          L104: basic chemical compo--- may be replaced by meat quality term?

  xii.          Table 1: change to phosphorus

xiii.          L151: killed may be replaced by slaughter

xiv.          In statistical analysis: Please make it clear that for slaughter age, 12 wk and 15 weeks for both sex combined? And for sex, age of both group combined; ? Authors may represent it better with not combining these data and analyzing in separate groups as at 12 week- male and female difference; at 15 weeks- male and female differences also for better clarity

  xv.          Results and discussion: need improvement and more specific

Comments on the Quality of English Language

Need moderate editing

Author Response

Comments made by Reviewer 3  and response to the comments.

Początek formularza

Comments and Suggestions for Authors

The present manuscript Kokoszyński et al evaluated the effect of slaughter age and sex  on the carcass and meat quality of pheasants. Authors have conducted a good and exhaustive study but still there is a lot of scope for improvement in each section of manuscript. Thus I recommend to work more on this manuscript so to provide a better impact of the study and wide readership. The manuscript also need to be checked for grammatical errors and sentencing.

After making responses to the reviewers' comments, the article was sent for revision of grammatical errors by a native speaker of MDPI English Language Editing Services recommended by the editor of Animals journal.

I have my observations as follows-

Comment 1

Simple summary: seems too much general; especially 23-26 lines

Answer

The other 2 reviewers did not make comments on the Simple Summary section The text was left unchanged. The text included in L23-27 provides basic information useful to the public about the bird species used in the experiment.

 Comment 2

 Abstract: please be more specific for traits as some traits may be confusing at L44

Answer

„affected the some basic chemical composition of the meat”

Changed to:

L47-49

It also significantly affected the protein and water contents of the pectoral and leg muscles, the intramuscular fat contents of the leg muscles,

Comment 3

 L 37: Please add further details on methodology such as 40 pheasants- plz mention the groups wise such as 20 male 20 female, age

Answer:

The study material consisted of 40 common pheasants.

Changed to:

L38-39

The study material consisted of 40 common pheasants, including 10 males and 10 females at 12 weeks of age and 10 males and 10 females at 15 weeks of age

Comment 4

L40: cutting work means plz?

Answer:

„Cutting work”

Changed to

L42-43

„work required to the cut the samples (cutting work)”

Comment 5

Keywords: Authors may follow like this

Pheasants, sex, age of slaughter, meat quality

Answer

Keywords changed to:

L57

pheasants, sex, age of slaughter, meat quality

Comment 6

L56: authors may use main 1or 2 references Orly

Answer

[1,2,3,4,5]

L65

Changed to

[1,2]

Renumbered for positions 3,4,5 are Biesiada-Drzazga (2011); Yamak et al. (2020); Kokoszyński et al. (2014)

Comment 7

 In introduction: Need to be more focussed on the objectives and hypothesis, further authors have mentioned the leg bone dimensions, but in introduction, this section is not covered.

Answer:

Added research hypothesis: the following text

L128-130

The research hypothesis assumed the influence of slaughter age and sex on the carcass and meat quality characteristics and leg bone traits of the examined farmed common pheasants.

L 110-120

Added:

In this study, the dimensions of the femur and tibia were also determined. For pheasants bred for introduction into the wild, anatomopathological bone characteristics, including those of leg bones, may partly determine their chances of survival. An excessive rate of weight gain as a result of an excessive concentration of nutrients in the diet or excessive feed intake by pheasants can lead to excessive stress on the femur and tibia, the occurrence of various diseases or pathological changes in the locomotor system, and restricted movement. In pheasants introduced into the wild, this reduces their chances of escaping predators and surviving. After the period of intensive growth in young pheasants, it is necessary to introduce a diet with a lower nutrient concentration, containing whole grains and other feedstuffs found in the wild, and to provide aviaries so that the pheasants are better prepared for life in the wild [2,13].

In a previous version of the paper, this text was placed in the "discussion" section due to a comment from one of the reviewers about the need to shorten the "Introduction" chapter

Comment 8

First two paragraphs may be merged and concised as the objective of the study is to see the effect of age and sex; so extensively describing the nutritive value seems not have much relevance here.

Answer
Deleted:

In a study by Straková et al. [10], the levels of most amino acids in the breast and leg muscles were higher than in broiler chicken muscles. According to , pheasant meat (breast or thigh)  a higher content of leucine, serine, tyrosine, arginine, glycine, alanine, valine, aspartic acid, and glutamic acid  broiler chicken meat.

It a high content of B vitamins, including vitamin B3 (niacin), B5 (pantothenic acid), B6 (pyridoxine) and B12 (cobalamin). Pheasant meat contains more phosphorus and magnesium and less calcium  broiler chicken meat [15]. It is also a good source of iron. Consumption of 100 g of drumstick meat covers 23.6% of the daily requirement for this element [6]. Pheasant meat also

Accordingly, a correction was made to the numbering of references in the main text.  

Comment 9 

L79-88: sorry for my observation, but it seems for me irrelevant and may be deleted

Answer:

Text left unchanged:

The article contains data on meat texture traits of pheasants. Data on meat texture traits (with the exception of shear force) have so far only been presented in articles by Balowski et al. (2015) and Kokoszynski et al. (2018). This section of the "Introduction" is closely related to the research results presented in this article.

Comment 10

L100-101: please correct the sentence also may add lack of studies or scarcity of published studies etc

Sentence „The authors are of any studies in the available literature” is correct.

was changed to:

The authors are not aware of any studies in the available literature on the effects of pheasant age……..

On the other hand, the phrase "No studies on..." often turns out to be untrue. Such research may have already been done and the results have not yet been published, or there are papers in the native language (for example, Arabic, Chinese or Japanese, Cyrillic, etc.) without a title and abstract in English or only in the printed version unavailable to the authors of this article.

Comment 11

L104: basic chemical compo--- may be replaced by meat quality term?

Answer

Of course you can but this paper presents only a small part of the research that falls under the term "meat quality"

Comment 12 

Table 1: change to phosphorus

Answer:

„hosphorus” changed to „Phosphorus

Comment 13

L151: killed may be replaced by slaughter

Answer
Killed manualny

Changed to

L178

„manually slaughtered”

Comment 14

In statistical analysis: Please make it clear that for slaughter age, 12 wk and 15 weeks for both sex combined? And for sex, age of both group combined; ? Authors may represent it better with not combining these data and analyzing in separate groups as at 12 week- male and female difference; at 15 weeks- male and female differences also for better clarity

Answer

One of the reviewers of the previous two rounds of reviews of this work found the layout of the presentation of main effects (12 wk, 15 wk, male, female) to be incorrect when there were any significant interactions. Therefore, at the request of this reviewer, the authors presented the results by gender (Male 12 wk, female 12 wk, Male 15 wk, female 15 wk). This arrangement is more insightful and more interesting for researchers or pheasant meat consumers. Therefore, the current way of presenting results by sex for a given pheasant evaluation date was left in place.  The occurrence of well-marked sexual dimorphism in pheasants also supports the presentation of results by sex at 12 and 15 weeks of age. 

In addition, the headings in Tables 2-7 have been corrected in accordance with the method of presentation.

Comment 15

Results and discussion: need improvement and more specific

The current way of presenting the results of this study, that is, by sex at 12 and 15 weeks of age of the pheasants, allowed a more detailed description of them.

Moreover in „Results” chapter

deleted (p < 0.05) or (p > 0.05)

L280 (15 weeks) changed to (12 and 15 weeks)

Corrected headers of tables 2 to 7

L338 added „The”

In the "Discussion" section, in turn, the sentences were added:     

added

L502-507

In this study, moreover, there was a significant effect of gender on the water and protein contents of the pectoral and leg muscles, as well as the intramuscular fat content of the leg muscles, which was not confirmed in the experiments by the cited authors [46]. On the other hand, Kuźniacka et al. [25] and Kokoszyński et al. [45] reported on the significant effect of sex on the protein contents of the pectoral and leg muscles of 16-week-old pheasants.

L482-484

The inferior tenderness of the pectoralis major muscles of the male pheasants compared to the females was probably mainly related to the larger diameter of the pectoralis major muscle fibers of the males compared to the females.

Text regarding bone dimensions has been removed.

In our study, the dimensions of the femur and tibia were also determined.

For pheasants bred for introduction into the wild, anatomopathological bone characteristics, including leg bones, may partly determine their chances of survival. Too rapid a rate of weight gain as a result of too high a concentration of nutrients in the diet or excessive feed intake by pheasants can lead to excessive stress on the femur and tibia, the occurrence of various diseases or pathological changes in the locomotor system, and restricted movement. In pheasants introduced into the wild, this reduces their chances of escaping predators and surviving. After the period of intensive growth of young pheasants, it is necessary to introduce a diet with a lower nutrient concentration, containing whole grains and other feedstuffs found in the wild, and to provide aviaries, so that the pheasants are better prepared for life in the wild.    

Round 2

Reviewer 2 Report (New Reviewer)

Comments and Suggestions for Authors

Thank you for your revisions 

Comments on the Quality of English Language

Minor editing 

Reviewer 3 Report (New Reviewer)

Comments and Suggestions for Authors

The manuscript is now improved and may be accepted for publication.

This manuscript is a resubmission of an earlier submission. The following is a list of the peer review reports and author responses from that submission.

Round 1

Reviewer 1 Report

Comments and Suggestions for Authors

Dear Authors,

Thank you for the corrections made. However, there are still some issues that need clarification.

Comment 1:

You have clarified that the pheasant breeding center reared approximately 900 birds, out of which 40 were selected. Since I am not an expert in pheasant breeding, I am unsure whether it is feasible to house all these birds in a single aviary, or if the center is equipped with smaller aviaries for keeping the birds. The initial description lacked information about the number of birds in the center, which you have now clarified. However, please describe whether the birds were kept in one large aviary or bred in smaller aviaries/pens. Although I understand that in this study a single bird is the basic experimental unit, as the experiment lacks experimental factors at the replicate levels such as different feeds, information on how these birds were randomly selected from all the birds in the center would be beneficial.

 Comment 2: I still believe that the introduction should be more focused on the study's aim. The information in lines 71-76 seems irrelevant and could be removed without affecting the manuscript's content. The shift to their attractiveness for hobby purposes distracts from the substantive flow of the introduction.

Comment 3: I still think that irrelevant information regarding bone function in birds should be removed. If you wish to describe the bone function in birds in relation to the aim of the manuscript, it should be done more succinctly. If you want to describe the bone function in pheasant in relation to the aim of the manuscript, there is a longish example of such a paragraph:

The present study also determined the dimensions of the femur and tibia of pheasants. In the case of pheasants bred for repopulation, anatomomorphological characteristics of their bones, as passive elements of the locomotor system, also partially determine their chances of survival. This applies not only to the wing bones, sternum, and shoulder girdle, whose quality determines the flying capabilities of the birds, but also to the long bones of the pelvic limb. The anatomical shape and size of the long bones determine their adaptation to counteracting gravitational forces and mechanical pressures that arise during movement activities. Due to the fact that breeding efforts are mainly focused on achieving the greatest possible muscle development in birds in breeding poultry, including pheasants, the growth rate of muscle tissue significantly exceeds the growth rate of the skeletal system. This might results in disproportionate growth of the pectoral muscles shifts the bird's center of gravity forward, disturbing the optimal distribution of body mass. The femur and tibia undergo excessive loads, leading to various types of diseases and pathological changes within the bones, resulting in impaired supportive and bearing functions of the entire locomotor system. This makes the birds struggle to maintain balance, move with difficulty, and tend to limit movement and adopt a sitting position. In birds reintroduced to the natural living environment, this reduces their chances of escaping from predators, while in birds reared for slaughter, it not only leads to a decrease in interest in consuming feed and water, but also results in deformations of the pectoral muscles.

 Comment 4: ok, thank you

Comment 5: ok, thank you

Comment 6: In original manuscript the sentence “Acidity (pH24) and electrical conductivity (EC24) of breast meat (m. Pectoralis major) were determined before carcass dissection and after cooling.” clearly indicates performing double measurements. The information that “after cooling” means 24h post-mortem was not provided.

 Comment 7: Thank you for this comment. However, as you see, the cited reference (Bourne 1982, or the second edition from 2022) clearly distinguishes between TPA and puncture tests. While during TPA tests gumminess, springiness, chewiness, and adhesiveness are determined, during the puncture test, as you wrote, maximum force, breaking strength, and penetration depth can be quantified. Further, Bourne was the first one who used compression with a probe larger than the sample in TPA tests (Bourne, M. 1968. Texture profile of ripening pears. J. Food Sci. 33, 223–226), indicating that during puncture a combination of compression and shear appears. Since then, back from 1968, he recommended using a „Probe to sample diameter ratio >1”, not <1 like in your study. In an article from 1998, Alina A. Szczesniak, the pioneer of TPA, who developed the original TPA parameters, wrote, “I am deeply perturbed by what I would call a ‘misuse’ of the method, and a poor understanding of the meaning of the parameters and the manner in which the method should be executed. As examples of ‘misuse,’ I would cite an article published not so long ago in the Journal of Texture Studies in which a penetrating needle (rather than a compressing plate) was used. […] Penetration causes a totally different structural damage than compression; penetrating the sample twice in the same spot leads to meaningless data and the test should not be called TPA.” Bourne’s reply was as follows: “I too have become concerned about misinterpretation and potential confusion from reports of measurements that purport to be TPA but have strayed from the original intent, procedure, and extracted data from the force-time curve.” (Bourne M., Szczesniak A.A. 1998, Issues pertaining to the texture profile analysis, J. Texture Studies, 29(4), vii-viii). In the TPA manual “Recommended TPA Test Method Guidelines” on the homepage of the USA distributor of Stable Micro Systems texturometers (https://texturetechnologies.com/resources/texture-profile-analysis), the one you used in your study, the following information can be found: “Occasionally it is acceptable to test products which are larger than the probe, such as sheet cakes, muffins, breads, with the understanding that if the probe breaks through and penetrates into the product then the resilience, springiness, and even cohesiveness values should be looked at skeptically (since the ability of the compressed product to spring back may be unduly negated by friction and sidewalls of the penetration hole).”

However, both notable TPA-pioneers (or cited guidelines) were not directly involved in (or do not directly refer to) the studies on meat or sausage products. Therefore, if Authors provide reliable references to studies of other researchers showing that using a “probe to sample diameter ratio < 1” is a commonly used and accepted procedure in TPA studies in the area of meat quality, this non-canonical quasi-TPA approach could be accepted.

Comment 8: ok, thank you

Comment 9: ok, thank you (https://www.fossanalytics.com/en/products/foodscan-2-meat-analyser)

Comment 10: ok, thank you

Comment 11: ok, thank you

Comment 12: ok, thank you

Comment 13: ok, thank you

Minor comments: ok, thank you

Author Response

Comments made by Reviewer No 1 and response to the comments.

DPoczątek formularza

ear Authors,

Thank you for the corrections made. However, there are still some issues that need clarification.

Comment 1:

You have clarified that the pheasant breeding center reared approximately 900 birds, out of which 40 were selected. Since I am not an expert in pheasant breeding, I am unsure whether it is feasible to house all these birds in a single aviary, or if the center is equipped with smaller aviaries for keeping the birds. The initial description lacked information about the number of birds in the center, which you have now clarified. However, please describe whether the birds were kept in one large aviary or bred in smaller aviaries/pens. Although I understand that in this study a single bird is the basic experimental unit, as the experiment lacks experimental factors at the replicate levels such as different feeds, information on how these birds were randomly selected from all the birds in the center would be beneficial.

Answer:

Text (previously L144-148)

“From 4 to 12 weeks inclusive, the pheasants were housed in a small 300 m2 aviary with a stocking rate of 3 birds/m2. In the last phase of rearing, i.e. from 13 to 15 weeks of age, the pheasants were kept in a large aviary, called the winter aviary. The 1000 m2 winter aviary housed 900 pheasants”

Changed to:

L127-134

From 4 to 9 weeks of age inclusive, the pheasants were housed in a small 300 m2 aviary                 (30 m × 10 m) at a stocking rate of 3 birds/m2.  In the last phase of rearing, that is, from 10 to 15 weeks of age, the pheasants were kept in two large aviaries, called winter aviaries. Each winter aviary had 6400 m2 (80 m × 80 m). Males were kept in one aviary and females in the other. Sex identification based on the colour of the feathers on the breast of the pheasant was performed at the end of nine weeks of age, during the transfer of the birds from the small aviary to the winter aviaries.

 Comment 2: I still believe that the introduction should be more focused on the study's aim. The information in lines 71-76 seems irrelevant and could be removed without affecting the manuscript's content. The shift to their attractiveness for hobby purposes distracts from the substantive flow of the introduction.

Answer

Deleted text in L71-76

Farm-raised pheasants also find their way into gardens and parks. Due to their attractive appearance, especially the multicolored plumage of mature males, they are a popular bird kept for hobby purposes. Live pheasants are also exported from Poland, primarily to Italy, Spain and France [1,2,3,4]. As an element of the natural environment, the pheasant enlivens the fishery, beautifies the area, is an ally of the farmer in the fight against weeds and insects [5].

Comment 3: I still think that irrelevant information regarding bone function in birds should be removed. If you wish to describe the bone function in birds in relation to the aim of the manuscript, it should be done more succinctly. If you want to describe the bone function in pheasant in relation to the aim of the manuscript, there is a longish example of such a paragraph:

The present study also determined the dimensions of the femur and tibia of pheasants. In the case of pheasants bred for repopulation, anatomomorphological characteristics of their bones, as passive elements of the locomotor system, also partially determine their chances of survival. This applies not only to the wing bones, sternum, and shoulder girdle, whose quality determines the flying capabilities of the birds, but also to the long bones of the pelvic limb. The anatomical shape and size of the long bones determine their adaptation to counteracting gravitational forces and mechanical pressures that arise during movement activities. Due to the fact that breeding efforts are mainly focused on achieving the greatest possible muscle development in birds in breeding poultry, including pheasants, the growth rate of muscle tissue significantly exceeds the growth rate of the skeletal system. This might results in disproportionate growth of the pectoral muscles shifts the bird's center of gravity forward, disturbing the optimal distribution of body mass. The femur and tibia undergo excessive loads, leading to various types of diseases and pathological changes within the bones, resulting in impaired supportive and bearing functions of the entire locomotor system. This makes the birds struggle to maintain balance, move with difficulty, and tend to limit movement and adopt a sitting position. In birds reintroduced to the natural living environment, this reduces their chances of escaping from predators, while in birds reared for slaughter, it not only leads to a decrease in interest in consuming feed and water, but also results in deformations of the pectoral muscles.

Answer:

Thank you for your comprehensive comment. An excerpt from the text of this commentary after modification was used in the Discussion section.

The text has been removed at the request of 2 reviewers:

L114-118 (previous version)

 The present study also determined the dimensions of the femur and tibia of pheasants. The bones of birds are passive elements of the locomotor system. They are a reservoir of calcium used in the formation of the eggshell and also influence the quality of poultry meat produced [25]. The bone marrow is involved in the formation of white (granulocytes, lymphocytes) and red (erythrocytes) blood cells [26].

Comment 4: ok, thank you

Answer: Thank you

Comment 5: ok, thank you

Answer: Thank you

Comment 6: In original manuscript the sentence “Acidity (pH24) and electrical conductivity (EC24) of breast meat (m. Pectoralis major) were determined before carcass dissection and after cooling.” clearly indicates performing double measurements. The information that “after cooling” means 24h post-mortem was not provided.

Answer:

L181

After cooling

Changed to:

L170

after cooling, 24 hours after slaughter.

Comment: pH and EC measurements were made only once, 24 hours after slaughter, after the carcass had cooled.

I apologise the information about measuring EC and pH twice was a mistake. I did not look at the Table while writing the text.

 Comment 7: Thank you for this comment. However, as you see, the cited reference (Bourne 1982, or the second edition from 2022) clearly distinguishes between TPA and puncture tests. While during TPA tests gumminess, springiness, chewiness, and adhesiveness are determined, during the puncture test, as you wrote, maximum force, breaking strength, and penetration depth can be quantified. Further, Bourne was the first one who used compression with a probe larger than the sample in TPA tests (Bourne, M. 1968. Texture profile of ripening pears. J. Food Sci. 33, 223–226), indicating that during puncture a combination of compression and shear appears. Since then, back from 1968, he recommended using a „Probe to sample diameter ratio >1”, not <1 like in your study. In an article from 1998, Alina A. Szczesniak, the pioneer of TPA, who developed the original TPA parameters, wrote, “I am deeply perturbed by what I would call a ‘misuse’ of the method, and a poor understanding of the meaning of the parameters and the manner in which the method should be executed. As examples of ‘misuse,’ I would cite an article published not so long ago in the Journal of Texture Studies in which a penetrating needle (rather than a compressing plate) was used. […] Penetration causes a totally different structural damage than compression; penetrating the sample twice in the same spot leads to meaningless data and the test should not be called TPA.” Bourne’s reply was as follows: “I too have become concerned about misinterpretation and potential confusion from reports of measurements that purport to be TPA but have strayed from the original intent, procedure, and extracted data from the force-time curve.” (Bourne M., Szczesniak A.A. 1998, Issues pertaining to the texture profile analysis, J. Texture Studies, 29(4), vii-viii). In the TPA manual “Recommended TPA Test Method Guidelines” on the homepage of the USA distributor of Stable Micro Systems texturometers (https://texturetechnologies.com/resources/texture-profile-analysis), the one you used in your study, the following information can be found: “Occasionally it is acceptable to test products which are larger than the probe, such as sheet cakes, muffins, breads, with the understanding that if the probe breaks through and penetrates into the product then the resilience, springiness, and even cohesiveness values should be looked at skeptically (since the ability of the compressed product to spring back may be unduly negated by friction and sidewalls of the penetration hole).”

However, both notable TPA-pioneers (or cited guidelines) were not directly involved in (or do not directly refer to) the studies on meat or sausage products. Therefore, if Authors provide reliable references to studies of other researchers showing that using a “probe to sample diameter ratio < 1” is a commonly used and accepted procedure in TPA studies in the area of meat quality, this non-canonical quasi-TPA approach could be accepted.

Answer:

We would like to thank the Reviewer for all valuable comments regarding the measurement methodology using the TPA test. We agree with the Reviewer that the ratio of sample size to pin/probe size is an important factor influencing the TPA test parameters. However, the research material used in the studies of different authors varied greatly. Assessment of the texture and rheological properties of meat and meat products is an analysis used according to this methodology, for example in the following works:

  • Żochowska J., Lachowicz K., Gajowiecki L., Sobczak M., Kotowicz M., Żych A. 2005. Effects of carcass weight and muscle on texture, structure and myofibre characteristic of wild boar meat. Meat Science, 71, 244-248.
  • Żochowska J., Lachowicz K., Gajowiecki L., Sobczak M., Kotowicz M., Żych A. 2006. Growth-related changes of muscle fibre characteristic and rheological properties of wild boars meat. Medycyna Weterynaryjna, 62, 47-50.
  • Żochowska-Kujawska J., Lachowicz K., Sobczak M., Gajowiecki L. 2006. Comparing texture, structure and myofibre of selected muscles of 2 species of fallow-deer. Animal Science, Supplement, 1, 16-17.
  • Żochowska-Kujawska J., Lachowicz K., Gajowiecki L., Sobczak M., Kotowicz M., Żych A. Mędrala D. 2007. Effects of massaging on hardness, rheological properties and structure of four wild boar muscles of different fibre type content. Meat Science, 75, 595-602.
  • Żochowska-Kujawska J., Lachowicz K., Sobczak M., Gajowiecki L., Oryl B. 2008. Effects of carcass weight and muscle on texture, structure, rheological properties and myofibre characteristics of deer. Medycyna Weterynaryjna, 64 (11), 1304-1307.
  • Sobczak M., Lachowicz K., Żochowska-Kujawska J. 2010. The influence of giant fibres on utility for production of massaged products of porcine muscle longissimus dorsi. Meat Science, 84, 4, 638-644.
  • Żochowska-Kujawska J., Lachowicz K., Sobczak M. 2010. Utility for production of massaged products of selected wild boar muscles originating from wetlands and arable area. Meat Science, 85, 461-466.
  • Żochowska-Kujawska J., Lachowicz K., Sobczak M. 2012. Effects of fibre type and kefir, wine lemon, and pineapple marinades on texture and sensory properties of wild boar and deer longissimus muscle. Meat Science, 92, 675-680.
  • Żochowska-Kujawska J., Lachowicz K., Sobczak M., Nędzarek A., Tórz A. 2013. Effects of natural plant tenderizers on proteolysis and texture of dry sausages produced with wild boar meat addition. African Journal of Biotechnology, 12(38), 5670-5677.
  • Żochowska-Kujawska J., Lachowicz K., Sobczak M., Bienkiewicz G., Tokarczyk G., Kotowicz M., Machcińska E. 2016. Compositional characteristics and nutritional quality of European beaver (Castor fiber L.) meat and their utility for sausage production. Czech Journal of Food Sciences, 34(1), 87-92.
  • Żochowska-Kujawska J. 2016. Effects of fibre type and structure of longissimus lumborum (LL), biceps femoris (BF) and semimembranosus (SM) deer muscles salting with different NaCl addition on proteolysis index and texture of dry-cured meats. Meat Science, 121, 390-396 .
  • Kokoszyński D., Saleh M., Bernacki Z., Kotowicz M., Sobczak M., Żochowska-Kujawska J., Stęczny K. 2018. Digestive tract morphometry and breast muscle microstructure in spent breeder ducks maintained in a conservation programme of genetic resources. Archiv fur Tierzuchtarchives of Animal Breeding, 61, 3, 373-378.
  • Żochowska-Kujawska J., Kotowicz M., Sobczak M., Lachowicz K., Wójcik J. 2019. Age-related changes in the carcass composition and meat quality of fallow deer (DAMA DAMA L.). Meat Science, 147, 37-43.
  • Żochowska-Kujawska J., M. Kotowicz, M. Sobczak, S. Lisiecki. Effect of Muscle Fibre Type on the Fatty Acids Profile and Lipid Oxidation of Dry-Cured Venison SM (semimembranosus) Muscle. Foods, 11, 14, 1-11.
  • Kokoszyński, B. Biesiada-Drzazga, J. Żochowska-Kujawska, M. Kotowicz, M. Sobczak, M. Saleh, M. Fik, H. Arpášová , C. Hrnčár, S. Kostenko. 2022. Effect of genotype and sex on carcase composition, physicochemical properties, texture and microstructure of meat from geese after four reproductive seasons. British Poultry Science, 63, 4, 519-527.

The use of the same measurement conditions each time allows for comparison of the influence of various factors on the mechanical properties of meat and meat products.

It is also worth mentioning that some pioneers of texture research were of the opinion that probe to sample diameter ratio (≤1) is a factor that directly influences the firmness values (Friedman et al., 1963).

Comment 8: ok, thank you

Answer: Thank you

Comment 9: ok, thank you (https://www.fossanalytics.com/en/products/foodscan-2-meat-analyser)

Answer: Thank you

Comment 10: ok, thank you

Answer: Thank you

Comment 11: ok, thank you

Answer: Thank you

Comment 12: ok, thank you

Answer: Thank you

Comment 13: ok, thank you

Answer: Thank you

Minor comments: ok, thank you

Answer: Thank you

The authors of this manuscript sincerely thank the reviewer for performing the review and for his valuable comments.

Dół formularza

©

Comments made by Reviewer No 1 and response to the comments.

DPoczątek formularza

ear Authors,

Thank you for the corrections made. However, there are still some issues that need clarification.

Comment 1:

You have clarified that the pheasant breeding center reared approximately 900 birds, out of which 40 were selected. Since I am not an expert in pheasant breeding, I am unsure whether it is feasible to house all these birds in a single aviary, or if the center is equipped with smaller aviaries for keeping the birds. The initial description lacked information about the number of birds in the center, which you have now clarified. However, please describe whether the birds were kept in one large aviary or bred in smaller aviaries/pens. Although I understand that in this study a single bird is the basic experimental unit, as the experiment lacks experimental factors at the replicate levels such as different feeds, information on how these birds were randomly selected from all the birds in the center would be beneficial.

Answer:

Text (previously L144-148)

“From 4 to 12 weeks inclusive, the pheasants were housed in a small 300 m2 aviary with a stocking rate of 3 birds/m2. In the last phase of rearing, i.e. from 13 to 15 weeks of age, the pheasants were kept in a large aviary, called the winter aviary. The 1000 m2 winter aviary housed 900 pheasants”

Changed to:

L127-134

From 4 to 9 weeks of age inclusive, the pheasants were housed in a small 300 m2 aviary                 (30 m × 10 m) at a stocking rate of 3 birds/m2.  In the last phase of rearing, that is, from 10 to 15 weeks of age, the pheasants were kept in two large aviaries, called winter aviaries. Each winter aviary had 6400 m2 (80 m × 80 m). Males were kept in one aviary and females in the other. Sex identification based on the colour of the feathers on the breast of the pheasant was performed at the end of nine weeks of age, during the transfer of the birds from the small aviary to the winter aviaries.

 Comment 2: I still believe that the introduction should be more focused on the study's aim. The information in lines 71-76 seems irrelevant and could be removed without affecting the manuscript's content. The shift to their attractiveness for hobby purposes distracts from the substantive flow of the introduction.

Answer

Deleted text in L71-76

Farm-raised pheasants also find their way into gardens and parks. Due to their attractive appearance, especially the multicolored plumage of mature males, they are a popular bird kept for hobby purposes. Live pheasants are also exported from Poland, primarily to Italy, Spain and France [1,2,3,4]. As an element of the natural environment, the pheasant enlivens the fishery, beautifies the area, is an ally of the farmer in the fight against weeds and insects [5].

Comment 3: I still think that irrelevant information regarding bone function in birds should be removed. If you wish to describe the bone function in birds in relation to the aim of the manuscript, it should be done more succinctly. If you want to describe the bone function in pheasant in relation to the aim of the manuscript, there is a longish example of such a paragraph:

The present study also determined the dimensions of the femur and tibia of pheasants. In the case of pheasants bred for repopulation, anatomomorphological characteristics of their bones, as passive elements of the locomotor system, also partially determine their chances of survival. This applies not only to the wing bones, sternum, and shoulder girdle, whose quality determines the flying capabilities of the birds, but also to the long bones of the pelvic limb. The anatomical shape and size of the long bones determine their adaptation to counteracting gravitational forces and mechanical pressures that arise during movement activities. Due to the fact that breeding efforts are mainly focused on achieving the greatest possible muscle development in birds in breeding poultry, including pheasants, the growth rate of muscle tissue significantly exceeds the growth rate of the skeletal system. This might results in disproportionate growth of the pectoral muscles shifts the bird's center of gravity forward, disturbing the optimal distribution of body mass. The femur and tibia undergo excessive loads, leading to various types of diseases and pathological changes within the bones, resulting in impaired supportive and bearing functions of the entire locomotor system. This makes the birds struggle to maintain balance, move with difficulty, and tend to limit movement and adopt a sitting position. In birds reintroduced to the natural living environment, this reduces their chances of escaping from predators, while in birds reared for slaughter, it not only leads to a decrease in interest in consuming feed and water, but also results in deformations of the pectoral muscles.

Answer:

Thank you for your comprehensive comment. An excerpt from the text of this commentary after modification was used in the Discussion section.

The text has been removed at the request of 2 reviewers:

L114-118 (previous version)

 The present study also determined the dimensions of the femur and tibia of pheasants. The bones of birds are passive elements of the locomotor system. They are a reservoir of calcium used in the formation of the eggshell and also influence the quality of poultry meat produced [25]. The bone marrow is involved in the formation of white (granulocytes, lymphocytes) and red (erythrocytes) blood cells [26].

Comment 4: ok, thank you

Answer: Thank you

Comment 5: ok, thank you

Answer: Thank you

Comment 6: In original manuscript the sentence “Acidity (pH24) and electrical conductivity (EC24) of breast meat (m. Pectoralis major) were determined before carcass dissection and after cooling.” clearly indicates performing double measurements. The information that “after cooling” means 24h post-mortem was not provided.

Answer:

L181

After cooling

Changed to:

L170

after cooling, 24 hours after slaughter.

Comment: pH and EC measurements were made only once, 24 hours after slaughter, after the carcass had cooled.

I apologise the information about measuring EC and pH twice was a mistake. I did not look at the Table while writing the text.

 Comment 7: Thank you for this comment. However, as you see, the cited reference (Bourne 1982, or the second edition from 2022) clearly distinguishes between TPA and puncture tests. While during TPA tests gumminess, springiness, chewiness, and adhesiveness are determined, during the puncture test, as you wrote, maximum force, breaking strength, and penetration depth can be quantified. Further, Bourne was the first one who used compression with a probe larger than the sample in TPA tests (Bourne, M. 1968. Texture profile of ripening pears. J. Food Sci. 33, 223–226), indicating that during puncture a combination of compression and shear appears. Since then, back from 1968, he recommended using a „Probe to sample diameter ratio >1”, not <1 like in your study. In an article from 1998, Alina A. Szczesniak, the pioneer of TPA, who developed the original TPA parameters, wrote, “I am deeply perturbed by what I would call a ‘misuse’ of the method, and a poor understanding of the meaning of the parameters and the manner in which the method should be executed. As examples of ‘misuse,’ I would cite an article published not so long ago in the Journal of Texture Studies in which a penetrating needle (rather than a compressing plate) was used. […] Penetration causes a totally different structural damage than compression; penetrating the sample twice in the same spot leads to meaningless data and the test should not be called TPA.” Bourne’s reply was as follows: “I too have become concerned about misinterpretation and potential confusion from reports of measurements that purport to be TPA but have strayed from the original intent, procedure, and extracted data from the force-time curve.” (Bourne M., Szczesniak A.A. 1998, Issues pertaining to the texture profile analysis, J. Texture Studies, 29(4), vii-viii). In the TPA manual “Recommended TPA Test Method Guidelines” on the homepage of the USA distributor of Stable Micro Systems texturometers (https://texturetechnologies.com/resources/texture-profile-analysis), the one you used in your study, the following information can be found: “Occasionally it is acceptable to test products which are larger than the probe, such as sheet cakes, muffins, breads, with the understanding that if the probe breaks through and penetrates into the product then the resilience, springiness, and even cohesiveness values should be looked at skeptically (since the ability of the compressed product to spring back may be unduly negated by friction and sidewalls of the penetration hole).”

However, both notable TPA-pioneers (or cited guidelines) were not directly involved in (or do not directly refer to) the studies on meat or sausage products. Therefore, if Authors provide reliable references to studies of other researchers showing that using a “probe to sample diameter ratio < 1” is a commonly used and accepted procedure in TPA studies in the area of meat quality, this non-canonical quasi-TPA approach could be accepted.

Answer:

We would like to thank the Reviewer for all valuable comments regarding the measurement methodology using the TPA test. We agree with the Reviewer that the ratio of sample size to pin/probe size is an important factor influencing the TPA test parameters. However, the research material used in the studies of different authors varied greatly. Assessment of the texture and rheological properties of meat and meat products is an analysis used according to this methodology, for example in the following works:

  • Żochowska J., Lachowicz K., Gajowiecki L., Sobczak M., Kotowicz M., Żych A. 2005. Effects of carcass weight and muscle on texture, structure and myofibre characteristic of wild boar meat. Meat Science, 71, 244-248.
  • Żochowska J., Lachowicz K., Gajowiecki L., Sobczak M., Kotowicz M., Żych A. 2006. Growth-related changes of muscle fibre characteristic and rheological properties of wild boars meat. Medycyna Weterynaryjna, 62, 47-50.
  • Żochowska-Kujawska J., Lachowicz K., Sobczak M., Gajowiecki L. 2006. Comparing texture, structure and myofibre of selected muscles of 2 species of fallow-deer. Animal Science, Supplement, 1, 16-17.
  • Żochowska-Kujawska J., Lachowicz K., Gajowiecki L., Sobczak M., Kotowicz M., Żych A. Mędrala D. 2007. Effects of massaging on hardness, rheological properties and structure of four wild boar muscles of different fibre type content. Meat Science, 75, 595-602.
  • Żochowska-Kujawska J., Lachowicz K., Sobczak M., Gajowiecki L., Oryl B. 2008. Effects of carcass weight and muscle on texture, structure, rheological properties and myofibre characteristics of deer. Medycyna Weterynaryjna, 64 (11), 1304-1307.
  • Sobczak M., Lachowicz K., Żochowska-Kujawska J. 2010. The influence of giant fibres on utility for production of massaged products of porcine muscle longissimus dorsi. Meat Science, 84, 4, 638-644.
  • Żochowska-Kujawska J., Lachowicz K., Sobczak M. 2010. Utility for production of massaged products of selected wild boar muscles originating from wetlands and arable area. Meat Science, 85, 461-466.
  • Żochowska-Kujawska J., Lachowicz K., Sobczak M. 2012. Effects of fibre type and kefir, wine lemon, and pineapple marinades on texture and sensory properties of wild boar and deer longissimus muscle. Meat Science, 92, 675-680.
  • Żochowska-Kujawska J., Lachowicz K., Sobczak M., Nędzarek A., Tórz A. 2013. Effects of natural plant tenderizers on proteolysis and texture of dry sausages produced with wild boar meat addition. African Journal of Biotechnology, 12(38), 5670-5677.
  • Żochowska-Kujawska J., Lachowicz K., Sobczak M., Bienkiewicz G., Tokarczyk G., Kotowicz M., Machcińska E. 2016. Compositional characteristics and nutritional quality of European beaver (Castor fiber L.) meat and their utility for sausage production. Czech Journal of Food Sciences, 34(1), 87-92.
  • Żochowska-Kujawska J. 2016. Effects of fibre type and structure of longissimus lumborum (LL), biceps femoris (BF) and semimembranosus (SM) deer muscles salting with different NaCl addition on proteolysis index and texture of dry-cured meats. Meat Science, 121, 390-396 .
  • Kokoszyński D., Saleh M., Bernacki Z., Kotowicz M., Sobczak M., Żochowska-Kujawska J., Stęczny K. 2018. Digestive tract morphometry and breast muscle microstructure in spent breeder ducks maintained in a conservation programme of genetic resources. Archiv fur Tierzuchtarchives of Animal Breeding, 61, 3, 373-378.
  • Żochowska-Kujawska J., Kotowicz M., Sobczak M., Lachowicz K., Wójcik J. 2019. Age-related changes in the carcass composition and meat quality of fallow deer (DAMA DAMA L.). Meat Science, 147, 37-43.
  • Żochowska-Kujawska J., M. Kotowicz, M. Sobczak, S. Lisiecki. Effect of Muscle Fibre Type on the Fatty Acids Profile and Lipid Oxidation of Dry-Cured Venison SM (semimembranosus) Muscle. Foods, 11, 14, 1-11.
  • Kokoszyński, B. Biesiada-Drzazga, J. Żochowska-Kujawska, M. Kotowicz, M. Sobczak, M. Saleh, M. Fik, H. Arpášová , C. Hrnčár, S. Kostenko. 2022. Effect of genotype and sex on carcase composition, physicochemical properties, texture and microstructure of meat from geese after four reproductive seasons. British Poultry Science, 63, 4, 519-527.

The use of the same measurement conditions each time allows for comparison of the influence of various factors on the mechanical properties of meat and meat products.

It is also worth mentioning that some pioneers of texture research were of the opinion that probe to sample diameter ratio (≤1) is a factor that directly influences the firmness values (Friedman et al., 1963).

Comment 8: ok, thank you

Answer: Thank you

Comment 9: ok, thank you (https://www.fossanalytics.com/en/products/foodscan-2-meat-analyser)

Answer: Thank you

Comment 10: ok, thank you

Answer: Thank you

Comment 11: ok, thank you

Answer: Thank you

Comment 12: ok, thank you

Answer: Thank you

Comment 13: ok, thank you

Answer: Thank you

Minor comments: ok, thank you

Answer: Thank you

The authors of this manuscript sincerely thank the reviewer for performing the review and for his valuable comments.

Reviewer 2 Report

Comments and Suggestions for Authors

This study evaluated the effect of slaughter age and sex on processing performance, meat quality, and bone characteristics of pheasants. The manuscript is fairly well written. However, the novelty of the study is limited. I have summarized my comments as follows:   

-When you raise all the birds together in the same space without any physical barriers (cages), Pseudoreplication occurs. Males and females should have been kept in separate cages, and each sex has sufficient replications.

-The Introduction section is long and contains unnecessary details. For example, the authors elaborated extensively on the nutritional value of pheasants and provided unnecessary details.

-For the statistical model (240), what does k stand for, and why b has ij? It isn’t the genotype by sex interaction (242). You are testing the slaughter age by sex interaction.

-The results that involved two-way interactions haven’t been interpreted correctly. The authors separated the main effect means and interpreted them while the p-value for interaction is significant, which is incorrect. If an interaction is significant, the factors do not act independently and interpretations should be based on simple effects and NOT the main effects.

-The result section lacks the live BW and the carcass as a % of the live WB. It is important in processing performance studies to show such data.

-Lines 458-461:  The results don’t strongly support your claim. As pheasant age, you get more meat (Table 2) and no texture characteristics are significantly different (Table 5).

-"Gender" refers to humans. "Sex" is a more appropriate term for animals.

Comments on the Quality of English Language

Just a few issues that need to be fixed. 

Author Response

Comments made by Reviewer No 2 and response to the comments

Open Review

Quality of English Language

( ) I am not qualified to assess the quality of English in this paper
( ) English very difficult to understand/incomprehensible
( ) Extensive editing of English language required
( ) Moderate editing of English language required
(x) Minor editing of English language required
( ) English language fine. No issues detected

Yes

Can be improved

Must be improved

Not applicable

Does the introduction provide sufficient background and include all relevant references?

( )

(x)

( )

( )

Are all the cited references relevant to the research?

( )

( )

( )

(x)

Is the research design appropriate?

( )

( )

(x)

( )

Are the methods adequately described?

( )

(x)

( )

( )

Are the results clearly presented?

( )

(x)

( )

( )

Are the conclusions supported by the results?

( )

( )

(x)

( )

Comments and Suggestions for Authors

This study evaluated the effect of slaughter age and sex on processing performance, meat quality, and bone characteristics of pheasants. The manuscript is fairly well written. However, the novelty of the study is limited. I have summarized my comments as follows:   

Comment 1

-When you raise all the birds together in the same space without any physical barriers (cages), Pseudoreplication occurs. Males and females should have been kept in separate cages, and each sex has sufficient replications.

Answer:

Text (previously L144-148)

“From 4 to 12 weeks inclusive, the pheasants were housed in a small 300 m2 aviary with a stocking rate of 3 birds/m2. In the last phase of rearing, i.e. from 13 to 15 weeks of age, the pheasants were kept in a large aviary, called the winter aviary. The 1000 m2 winter aviary housed 900 pheasants”

Changed to:

L127-134

From 4 to 9 weeks of age inclusive, the pheasants were housed in a small 300 m2 aviary                (30 m × 10 m) at a stocking rate of 3 birds/m2.  In the last phase of rearing, that is, from 10 to 15 weeks of age, the pheasants were kept in two large aviaries, called winter aviaries. Each winter aviary had 6400 m2 (80 m × 80 m). Males were kept in one aviary and females in the other. Sex identification based on the colour of the feathers on the breast of the pheasant was performed at the end of nine weeks of age, during the transfer of the birds from the small aviary to the winter aviaries.

This was due to the verification of information previously provided by the manager of the pheasant farm. Mr Maciej Imański has only been in charge of the Polish Hunting Association's pheasant farm in Rożniaty for a few months, so some initial data were checked (dimensions and number of aviaries) and corrected.

Comment 2

-The Introduction section is long and contains unnecessary details. For example, the authors elaborated extensively on the nutritional value of pheasants and provided unnecessary details.

Answer:

Deleted:

L71-77 (previously)

Farm-raised pheasants also find their way into gardens and parks. Due to their attractive appearance, especially the multicolored plumage of mature males, they are a popular bird kept for hobby purposes. Live pheasants are also exported from Poland, primarily to Italy, Spain and France [1,2,3,4]. As an element of the natural environment, the pheasant enlivens the fishery, beautifies the area, is an ally of the farmer in the fight against weeds and insects [5].

L114-118

The present study also determined the dimensions of the femur and tibia of pheasants. The bones of birds are passive elements of the locomotor system. They are a reservoir of calcium used in the formation of the eggshell and also influence the quality of poultry meat produced [25]. The bone marrow is involved in the formation of white (granulocytes, lymphocytes) and red (erythrocytes) blood cells [26].

The text about nutritional value has been left out because the meat quality parameters are presented in this article.

Comment 3

-For the statistical model (240), what does k stand for, and why b has ij? It isn’t the genotype by sex interaction (242). You are testing the slaughter age by sex interaction.

-The results that involved two-way interactions haven’t been interpreted correctly. The authors separated the main effect means and interpreted them while the p-value for interaction is significant, which is incorrect. If an interaction is significant, the factors do not act independently and interpretations should be based on simple effects and NOT the main effects.

Answer:

bji changed to bj (L236)

L241-242 added

k- is the kth observation for the target trait in ij group.

Other:

This article presents the results of the study for the main factors (slaughter age, sex) without presenting data for male and female pheasants at 12 and 15 weeks of age. This way of presenting research results is often presented in other previous scientific studies. The other reviewers did not comment on this way of presenting the results of this study.

Comment 4

-The result section lacks the live BW and the carcass as a % of the live WB. It is important in processing performance studies to show such data.

Answer:

Information about body weight and dressing percentage was not presented in this article due to the fact that the body weight of pheasants at 12 weeks of age determined on the farm using a dial scale with an accuracy of 50 g differed little from the carcass weight determined on the electronic laboratory scale. The calculated yield was most often between 85 and 87%, much higher than in previous studies. We decided not to report these data as unreliable. Furthermore, we do not have data on the BW of pheasants at 15 weeks of age.

Comment 5

-Lines 458-461:  The results don’t strongly support your claim. As pheasant age, you get more meat (Table 2) and no texture characteristics are significantly different (Table 5).

The difference between the pectoral muscle weight of males and females was 67.5 g (males 211.4 g, females 143.9 g), while the difference in BW between pheasants of both sexes at               12 and 15 weeks of age was 41.7 g (12 wk 157.1 g. 15 wk 198.8 g). Probably for these reasons, it was sex and not age that significantly differentiated the birds studied in terms of texture characteristics.

Comment 6

-"Gender" refers to humans. "Sex" is a more appropriate term for animals.

 Answer:

„Gender” changed to „sex” in the whole main text.

Comments on the Quality of English Language

Just a few issues that need to be fixed. 

Answer:

After proofreading the article according to the comments of the three main reviewers, the text was checked by one person — a language professional with expertise in scientific writing.

The authors of this manuscript sincerely thank the reviewer for performing the review and for his valuable comments.

Reviewer 3 Report

Comments and Suggestions for Authors

Dear Authors,
Please find your comments on the attachment.

Comments on the Quality of English Language

The manuscript has not been presented in an intelligent manner. There are grammatical and general writing mistakes in the manuscript. I recommend revision of the manuscript by a native English speaker or language professional with expertise in scientific writing. 

Author Response

Comments made by Reviewer No 3 and response to the comments

OPoczątek formularza

pen Review

Quality of English Language

( ) I am not qualified to assess the quality of English in this paper
( ) English very difficult to understand/incomprehensible
( ) Extensive editing of English language required
(x) Moderate editing of English language required
( ) Minor editing of English language required
( ) English language fine. No issues detected

Yes

Can be improved

Must be improved

Not applicable

Does the introduction provide sufficient background and include all relevant references?

( )

( )

(x)

( )

Are all the cited references relevant to the research?

(x)

( )

( )

( )

Is the research design appropriate?

(x)

( )

( )

( )

Are the methods adequately described?

( )

(x)

( )

( )

Are the results clearly presented?

( )

(x)

( )

( )

Are the conclusions supported by the results?

(x)

( )

( )

( )

Comments and Suggestions for Authors

Dear Authors,
Please find your comments on the attachment.

Comments on the Quality of English Language

The manuscript has not been presented in an intelligent manner. There are grammatical and general writing mistakes in the manuscript. I recommend revision of the manuscript by a native English speaker or language professional with expertise in scientific writing. 

Answer:

After proofreading the article according to the comments of the three main reviewers, the text was checked by one person — a language professional with expertise in scientific writing.

Dół formularza

© 1996-2024 MDPI (Basel, Switzerland) unless otherwise stated

Comments provided in PDF version

General Comments This manuscript discusses some of the effects of age and sex on carcass characteristics and bone dimensions in pheasants.

Overall, this is a well-designed study. However, the authors should make some changes, including the

following:

Comment 1

The abstract is lengthy and has a word count more than double, as recommended by the journal. A fully structured abstract within the word limit should be written.

Answer:

The Abstract chapter was increased by 99 words at the request of the reviewers in Round 1 of the article review. It has now been reverted to its original state.

Deleted:

The undertaken research made it possible to identify the appropriate slaughter age in relation to consumer requirements and the differences between male and female pheasants in terms of carcass composition and meat quality. The results of these studies also provided information on the dimensions of the femur and tibia bones of males and females of the common pheasant and the changes in these leg bone dimensions occurring with the age of the birds.

In this study, we found a significant effect of both factors (slaughter age and sex) on the nutritional value and suitability for processing of pheasant meat.

Comment 2

More emphasis should be placed on the parameters that are gaps in existing studies. I recommend revising the Introduction.

Deleted:

L71-77 (previously)

Farm-raised pheasants also find their way into gardens and parks. Due to their attractive appearance, especially the multicolored plumage of mature males, they are a popular bird kept for hobby purposes. Live pheasants are also exported from Poland, primarily to Italy, Spain and France [1,2,3,4]. As an element of the natural environment, the pheasant enlivens the fishery, beautifies the area, is an ally of the farmer in the fight against weeds and insects [5].

L114-118 (previously)

The present study also determined the dimensions of the femur and tibia of pheasants. The bones of birds are passive elements of the locomotor system. They are a reservoir of calcium used in the formation of the eggshell and also influence the quality of poultry meat produced [25]. The bone marrow is involved in the formation of white (granulocytes, lymphocytes) and red (erythrocytes) blood cells [26].

Comment 3

Particularly, focus on the research gap, which has already been discussed but needs more comprehensive development of the gap, hypothesis, and connection with the aim of the study.

Answer

Added:

L456-459

The texture of meat is an attribute of its quality, affecting acceptance, desirability, eating experience and consumer satisfaction. Characteristics such as tenderness, juiciness, and chewiness have a strong influence on consumer perception of meat as a food product [45].

L471-485

In our study, the dimensions of the femur and tibia were also determined. Only information on the length of the tibia bone of pheasants was found in the available literature. In the experiment of Flis et al. [47], the length of the tibia bone of 16-week-old male game pheasants ranged from 108.4 to 114.4 mm, i.e. close to the greatest length of the tibia bone of the pheasants in this study. For pheasants bred for introduction into the wild, anatomopathological bone characteristics, including leg bones, may partly determine their chances of survival. Too rapid a rate of weight gain as a result of too high a concentration of nutrients in the diet or excessive feed intake by pheasants can lead to excessive stress on the femur and tibia, the occurrence of various diseases or pathological changes in the locomotor system and restricted movement. In pheasants introduced into the wild, this reduces their chances of escaping predators and surviving. After the period of intensive growth of young pheasants, it is necessary to introduce a diet with a lower nutrient concentration, containing whole grains and other feedstuffs found in the wild, and to provide aviaries with considerably more space, so that the pheasants are better prepared for life in the wild.

Comment 4

The manuscript has not been presented in an intelligent manner. There are grammatical and general writing mistakes in the manuscript. I recommend revision of the manuscript by a native English speaker or language professional with expertise in scientific writing.

Answer:

After proofreading the article according to the comments of the three main reviewers, the text was checked by one person - a language professional with expertise in scientific writing.

Specific Comments

Comment 5

There are a few specific suggestions that might improve the manuscript: Line 2: gender? Sex is the most appropriate word to be added. Check the complete manuscript to replace gender with sex.

Answer:

Word „gender” changed to „sex” in the whole main text of this artice.

Comment 6

 Lines 7-22: Affiliation of the authors and their initials must be re-checked and corrected as per journal guidelines, particularly for corresponding authors.

Answer:

L7-22

Postcode numbers and initials of correspondent authors have been added.

Comment 7

 Line 33: Nutritional values

Answer:

L33

nutritional value changed to nutritional values, added „s”

Comment 8

 Line 42: maiz grains were additionally introduced into the diet. Also mention the amount used on daily basis.

Answer

L42, 142

Added „(60 g/bird/day)”

Comment 9

 Line 45: breast meat, and protein, ….

Answer

L45-46

breast meat, as well as protein

Changed to:

breast meat and protein

Comment 10

Line 52: tibia bone dimensions were assessed

Answer:

Line 52-53: tibia bone dimensions  assessed.

Changed to:

Line 52: tibia bone dimensions were assessed.

Comment 11

Lines 52-53: This study

Answer:

Sentence deleted at the request of Reviewers 1 and 2

Comment 12

Line 58: We confirmed? Rewrite the last sentence in the abstract.

Answer:

Sentence deleted at the request of Reviewers 1

The original text of the Abstract chapter has been reverted. Chapter text reduced by 99 words.

Comment 13

Line 69: In Poland, pheasant carcasses used for consumption

Answer:

In Poland, pheasant carcasses for consumption

Changed to:

Line 63: In Poland, pheasant carcasses used for consumption

Comment 14

Line 81: Replace “while” by “whereas”

Answer:

L 70

„while” changed to „whereas”

Comment 15

Line 101: “in order to obtain” by “to obtain light carcass that is well suited…”

Answer:

L101

„in order to obtain light carcass well suited to”

Changed to

L91

„to obtain light carcass that is well suited to”

Comment 16

Line 104: “On the other hand” to “In contrast”

Answer:

L104

“On the other hand”

Changed to:

L94

 “In contrast”

Comment 17

Line 119: and tibial and femoral bone dimensions.

Answer:

L119

and tibia and femor bone dimensions

changed to

L104-105

and tibial and femural bone dimensions

Comment 18

Line 129: Birds ages can be written in parenthesis.

Answer:

Left unchanged. The notation with age in brackets was incomprehensible to other reviewers in round 1 of the article review.

Comment 19

Line 142: from 4 to 12 weeks of age….

Answer:

L142

from 4 to 12 weeks….

Changed to:

L127

From 4 to 9 weeks of age…. (correction of the deadline)

Comment 20

Line 143: 300 m2 aviary at a stocking rate of 3 birds/m2 .

Answer

L143

Sentence „300 m2 aviary with a shocking rate of 3 birds/m2

Changed to:

L128-129

„300 m2 aviary (30 m × 10 m) at a stocking rate of 3 birds/m2

Comment 21

Line 144: In the last phase of rearing, that is, from 13 to 15 weeks of age….

Answer

L144

In the last phase of rearing, i.e. from 13 to 15 weeks of age..

Changed to:

L129

In the last phase of rearing, that is, from 10 to 15 weeks of age….

Comment

A correction to the deadline was also made on the basis of verified information from the manager of the Rożniaty PZŁ farm, Mr Maciej Imański. Mr Maciej Imański has only been the manager of the pheasant farm for a few months - he is in the process of familiarising himself with pheasant farming.

 Comment 22

Line 152: Replace “i.e.” with “that is”

Answer:

L140 „i.e.’ changed to „that is”

Comment 23

Line 166-167: Correct the capitalization error, and what were the selection criteria out of the 900 birds?

Answer:

Pheasant Breeding Center – is a proper name, hence the capital letters (L155)

L155-157  added:

In each aviary (males and females were kept in separate winter aviaries from week 9 onwards), birds were obtained at several remote locations.

Comment 24

Lines 168-169: Rewrite the slaughter procedure.

Answer:

The birds were slaughtered manually (bludgeoning to stun, cutting the neck veins, bleeding).

Changed to:

L158-159

The pheasants were manually killed by stunning with a club and bled through a ventral cut of the neck blood vessels.

Comment 25

Line 178: Acidity (pH24 hours)?? pH24 h may be the most appropriate term.

Answer:

L169

(pH24 hours) changed to (pH24 h)

(EC24 hours) changed to (EC24 h)

Comment 26

Line 183: m. pectoralis major or M. cectoralis major

Answer:

L169

  1. Pectoralis major changed to: m. pectoralis major

Comment 27

Lines 202-203: The pH24 measurements were repeated three times for each sample.

Answer

The pH24 measurements was repeated 3 times for each sample.

Changed to:

L194-195

The pH24 measurements were repeated three times for each sample.

Comment 28

 Line 204: Breat or leg muscle? Provide a detailed procedure.

Answer

L198-199

L206 added (protein, intramuscular fat, water and collage contents) after chemical composition

L199

breast or leg muscle changed breast and leg muscle

L201 added „separately” after determined

Comment 29

Line 244: SAS software

Answer:

SAS computer program

Changed to

L242

SAS software

Comment 30

Line 254: more skin

Answer

L251

higher skin changed to „more skin”

Comment 31

Line 284: breast muscle or leg muscle?? Please explain

Answer

L284

„or” changed to „and”,

it was about the separate designation of breast muscle and leg muscle

Comment 32

Line 360: [21] analyzed….

Answer

L378

„analizing” changed to „analyzed”

Comment 33

Line 362: (18 weeks, 4.3%; 20 weeks, 4.2%)

Answer:

(18 wks – 4.3%, 20 wks – 4.2%)

Changed to

L380-381

(18 weeks, 4.3%; 20 weeks , 4,2%)

Comment 34

Line 397: However, to date,……

Answer

So far, however

Changed to

L471

However, to date,

Comment 35

Lines 400-401: According to me there is no need to add these lines to support the discussion. Line 402: Important is sufficient. No need to add very.

Answer:

Deleted 

L402-403 (previously)

In studies conducted on broilers [40] or ducks [41], lower EC24 breast meat values were obtained.

L421 „very” important changed to important

Comment 36

Lines 240-243: Equations and their explanation must be written and formatted as per journal instruction

L237-282

Equations and their explanation were done according to the journal instructions

Comment 37

Line 455: than those of 12-week-old birds.

Answer:

„than 12-week-old birds”

Changed to

L489

than those of 12-week-old birds.

Comment 38

Line 260: between of?Line 464: lightness (L*) and redness (a*) of breast meat, and Tables: Recheck and correct the table numbers within the text. Table 1. Metabolizable energy (kcal/kg); Crude fat (%), correct the value of pheasant 1. Nutrient composition is missing. The chemical composition of the diets is analyzed or calculated composition? If calculated, the analyzed composition must be provided. Tables 2-5: p-value; Sex (S) and must be presented with separate cells not with age. Abbreviations used within the tables must be provided in full form in the table footnotes, irrespective of full form in the main text of manuscript.

Answer:

L260 between of – is correct

Table 1 has been corrected

For Pheasant 1 – for Metabolizable energy „(kcal/kg)” instead of „kcal on kg of feed”

and „3.00” instead of „.3.00” for crude fat

In the text, the numbering of Tables 2 to 7 has been corrected

Other:

The energy value, basic chemical composition, amino acid and mineral content are calculated values. As I mentioned in the 1st round of responses to the reviews, Wytwórnia Pasznia "Lira" Sp. z o.o. from Krzywiń (Wielkopolskie Voivodeship) Poland - producer of feed mixes Pheaasnt 1, Pheasant 2, Pheasant 3 did not agree to make available and publish the recipes of the feed mixes (trade secret). As in other Compound Feed Factories, the chemical composition of the feeds from which the mixtures are produced is determined, then a computerised recipe is laid out taking into account the values given from the feeding standards. Finally, the data are made available to pheasant producers.

Comment 39

Table 5: Cohesiviness? Unit?

Answer:

The attribute 'Cohesiviness' has no unit of measurement.

The authors of this manuscript sincerely thank the reviewer for performing the review and for his valuable comments.

Round 2

Reviewer 1 Report

Comments and Suggestions for Authors

Dear Authors,

I have requested that the Authors provide references to studies by other researchers demonstrating the use of a "probe to sample diameter ratio < 1." In response, I received a list of their own previous work. The assertion that they have been incorrectly performing TPA tests for nearly 20 years is not a sufficient reason to accept this fact and retract my critical remarks. Moreover, the cited work by Friedman et al., 1963, if it refers to the paper by Friedman, Whitney, & Szczesniak (10.1111/j.1365-2621.1963.tb00216.x), does not address firmness values at all. On the contrary, it clearly states that during the TPA test “the food sample has an area at least that of the plunger base” (p.393), indicating that a probe to sample diameter ratio ≤1 cannot be applied. Furthermore, the sentence quoted by the authors from the work by Ibanez et al (2022) (doi: 10.1111/1541-4337.12957) only indicates that a probe to sample diameter ratio <1 is a factor that directly influences the results, with which I cannot agree more and about which I mentioned previously, and which we try to avoid by using a ratio >1.

Therefore, I reiterate my request for references to works in the area of meat science by other authors where a probe to sample diameter ratio was <1 and for an indication in the paper that the texture measurement test performed is not a TPA test in its classic understanding.

Reviewer 2 Report

Comments and Suggestions for Authors

Not all my comments are addressed. Comments #1 and 3 are still unanswered. 

Reviewer 3 Report

Comments and Suggestions for Authors

Dear Authors,
Thank you for your great efforts to resubmit and answer my questions and concerns. I have evaluated the manuscript and regret to inform you that I am still unable to recommend it for publication at this time due to several concerns in the manuscript, including the following:

1. The recommendations regarding the Abstract and Introduction have not been fully addressed, and this section requires further improvement.

2. The writing of the manuscript still does not follow the scientific writing style, there is a gap in connecting the different sections with the next sections, and the manuscript lacks coherence in the writing. The results can be discussed and concluded in a much better manner.

3. The language of this manuscript requires further improvement.

Given these concerns, I am unable to recommend this manuscript for consideration at this stage.

Comments on the Quality of English Language

The language of this manuscript requires further improvement.